# Three-dimensional hidden phase probed by in-plane magnetotransport in kagome metal $CsV_3Sb_5$ thin flakes

Xinjian Wei[1,11], Congkuan Tian[1,2,11], Hang Cui[2], Yuxin Zhai[3], Yongkai Li[4,5,6], Shaobo Liu[1,2], Yuanjun Song[1], Ya Feng[1], Miaoling Huang[1], Zhiwei Wang [4,5,6], Yi Liu[7], Qihua Xiong [1,3], Yugui Yao [4,5,6], X. C. Xie[2,8,9] & Jian-Hao Chen [1,2,8,10] ✉

Transition metal compounds with kagome structure have been found to exhibit a variety of exotic structural, electronic, and magnetic orders. These orders are competing with energies very close to each other, resulting in complex phase transitions. Some of the phases are easily observable, such as the charge density wave (CDW) and the superconducting phase, while others are more challenging to identify and characterize. Here we present magneto-transport evidence of a new phase below ~ 35 K in the kagome topological metal $CsV_3Sb_5$ (CVS) thin flakes between the CDW and the superconducting transition temperatures. This phase is characterized by six-fold rotational symmetry in the in-plane magnetoresistance (MR) and is connected to the orbital current order in CVS. Furthermore, the phase is characterized by a large in-plane negative magnetoresistance, which suggests the existence of a three-dimensional, magnetic field-tunable orbital current ordered phase. Our results highlight the potential of magneto-transport to reveal the interactions between exotic quantum states of matter and to uncover the symmetry of such hidden phases.

The recently discovered kagome topological metal $AV_3Sb_5$ (A = Cs, Rb, K) has proven to be a valuable material platform for studying topological states and electron correlations[1–8]. It features a wealth of states of matter and interesting electronic behaviors, including topological surface states[2,9], superconductivity with pair density wave[3], electronic nematicity[8], charge density wave[4], chiral transport[10], anomalous Hall effect[11] and time-reversal symmetry breaking[7], among others. Such an intricate and diverse range of states has sparked great interest, and numerous experiments are quickly focused on the search for

potentially impactful quantum states within this system, such as unconventional superconductivity[4,12–14], Majorana zero mode[5,15], and orbital current order[16–22].

Taking $CsV_3Sb_5$(CVS) as an example, the two most visible phase transitions are the CDW transition[2,23] at around 90 K and the superconductivity transition[2,23] at around 2.5 K. Interestingly, an increasing number of experiments have suggested the presence of additional phase transitions between these two temperatures, with one potential transition at approximately 35 K. Muon spin-rotation (μSR) experiments

[1]Beijing Academy of Quantum Information Sciences, Beijing, China. [2]International Center for Quantum Materials, School of Physics, Peking University, Beijing, China. [3]State Key Laboratory of Low-Dimensional Quantum Physics and Department of Physics, Tsinghua University, Beijing, China. [4]Centre for Quantum Physics, Key Laboratory of Advanced Optoelectronic Quantum Architecture and Measurement, School of Physics, Beijing Institute of Technology, Beijing, China. [5]Beijing Key Lab of Nanophotonics and Ultrafine Optoelectronic Systems, Beijing Institute of Technology, Beijing, China. [6]Material Science Center, Yangtze Delta Region Academy of Beijing Institute of Technology, Jiaxing, China. [7]Center for Advanced Quantum Studies and Department of Physics, Beijing Normal University, Beijing, China. [8]Hefei National Laboratory, Hefei, China. [9]Institute for Nanoelectronic Devices and Quantum Computing, Fudan University, Shanghai, China. [10]Key Laboratory for the Physics and Chemistry of Nanodevices, Peking University, Beijing, China. [11]These authors contributed equally: Xinjian Wei, Congkuan Tian. ✉e-mail: chenjianhao@pku.edu.cn

showed a sudden increase in the relaxation rate below ~ 35 K[24,25]; STM, nuclear magnetic resonance (NMR), and elastoresistance measurement (EM) have pointed to the formation of electronic nematic order below ~ 35 K[8]; A second-harmonic generation (SHG) experiment found prominent chirality along the out-of-plane direction emerges below ~ 35 K[10]; Meanwhile, another STM experiment[26] found that the unidirectional coherent quasiparticles appear below 30 K. These studies altogether presented a puzzling physical picture, that the hidden phase below ~ 35 K simultaneously breaks the rotational symmetry and time-reversal symmetry. Moreover, its mechanism become more confusing since different conclusions have been reported recently, that spontaneously time-reversal symmetry breaking either coincides with CDW[27–29] or does not occur at all[30,31], and rotational symmetry breaking also occurs at higher temperatures[26–28]. With the limited number of experimental findings, much remains unknown about this hidden phase, including the exact mechanism that breaks time-reversal symmetry, the spatial symmetry of the order, and its magnetotransport characteristics.

In this study, we investigate the in-plane magnetoresistance of CVS thin flakes to understand the impact of magnetic fields and temperature on its electronic symmetry breaking. Our findings reveal that the hidden phase in CVS below ~ 35 K has a unique in-plane symmetry that is tunable by magnetic fields and is accompanied by substantial in-plane negative magnetoresistance. This transport result supports the possibility of a three-dimensional orbital current order which emerges below ~ 35 K with strong interlayer interactions. Our findings offer further insights into the microscopic mechanism underlying the hidden orders in CVS.

## Results and discussion

The schematic diagram of the in-plane MR measurement in this study is depicted in Fig. 1a. In the diagram, the current flows along the $x$-direction, while the $y$-axis is the in-plane direction perpendicular to $x$, and the $z$-axis is perpendicular to the atomic layers of the CVS crystal. The magnetic field is applied in the $x$-$y$ plane at an angle of $\gamma$ with

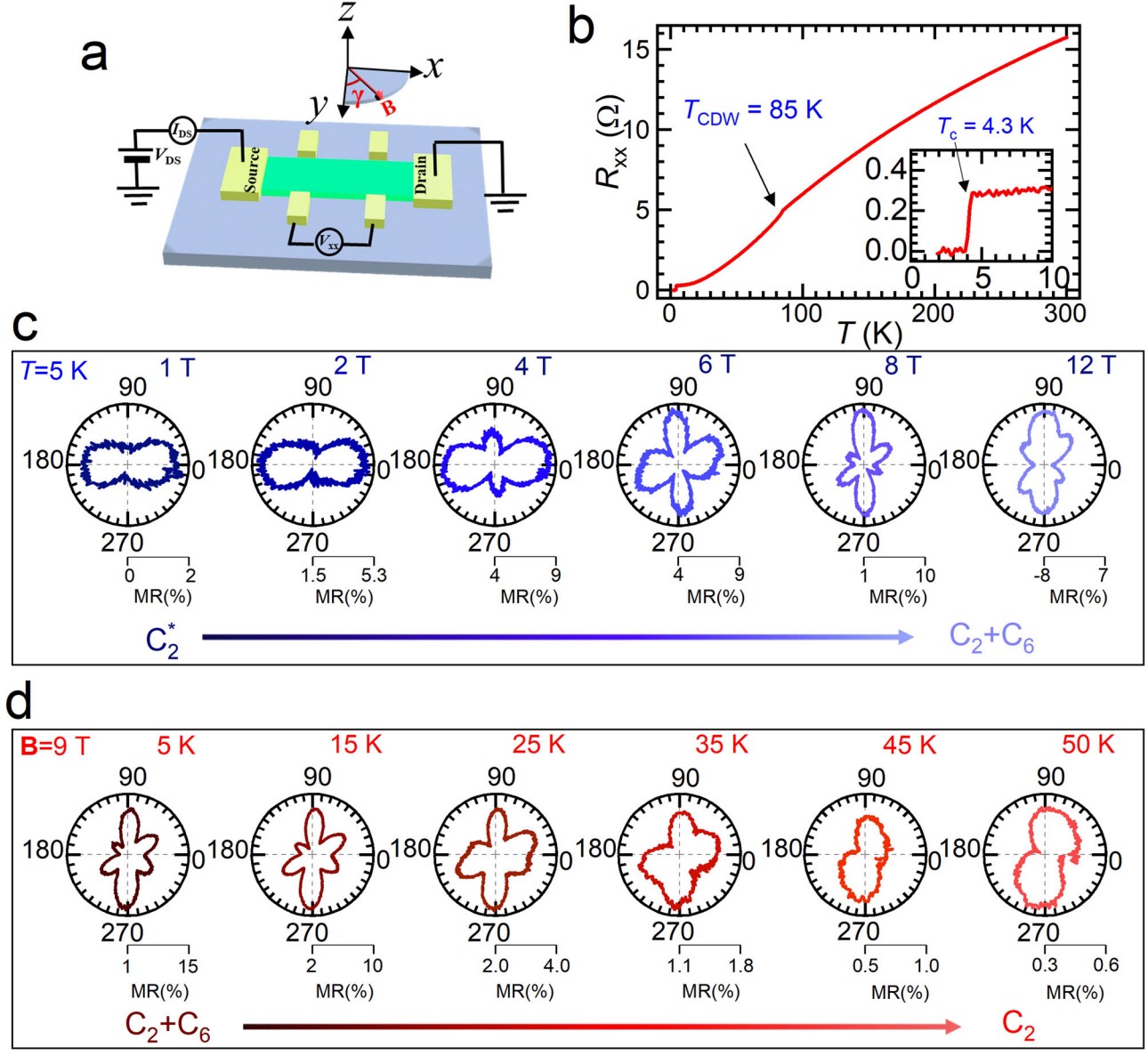

**Fig. 1 | In-plane magnetoresistance measurement of CVS crystal. a** Schematic diagram of in-plane magnetoresistance measurement. Here $\gamma$ is the angle between the magnetic field direction and the $y$ axis in the $x$-$y$ plane. **b** Resistance vs. temperature ($R$-$T$) curve at zero magnetic fields of a typical CVS sample of 32 nm thick and for $T$ ranging from 2 K to 300 K. The CDW transition can be seen as a kink in the $R$-$T$ curve at 85 K. Lower inset: zoom-in $R$-$T$ curve between 2 K and 10 K, showing a sharp superconductivity transition at 4.3 K. **c**, **d** Polar representation of the in-plane MR vs. $\gamma$ at $T$ = 5 K for **B** = 1, 2, 4, 6, 8, 12 T in Fig. 1c and at **B** = 9 T for $T$ = 5, 15, 25, 35, 45, 50 K in Fig. 1d. The center point in each polar plot is offset to clearly show the MR anisotropy.

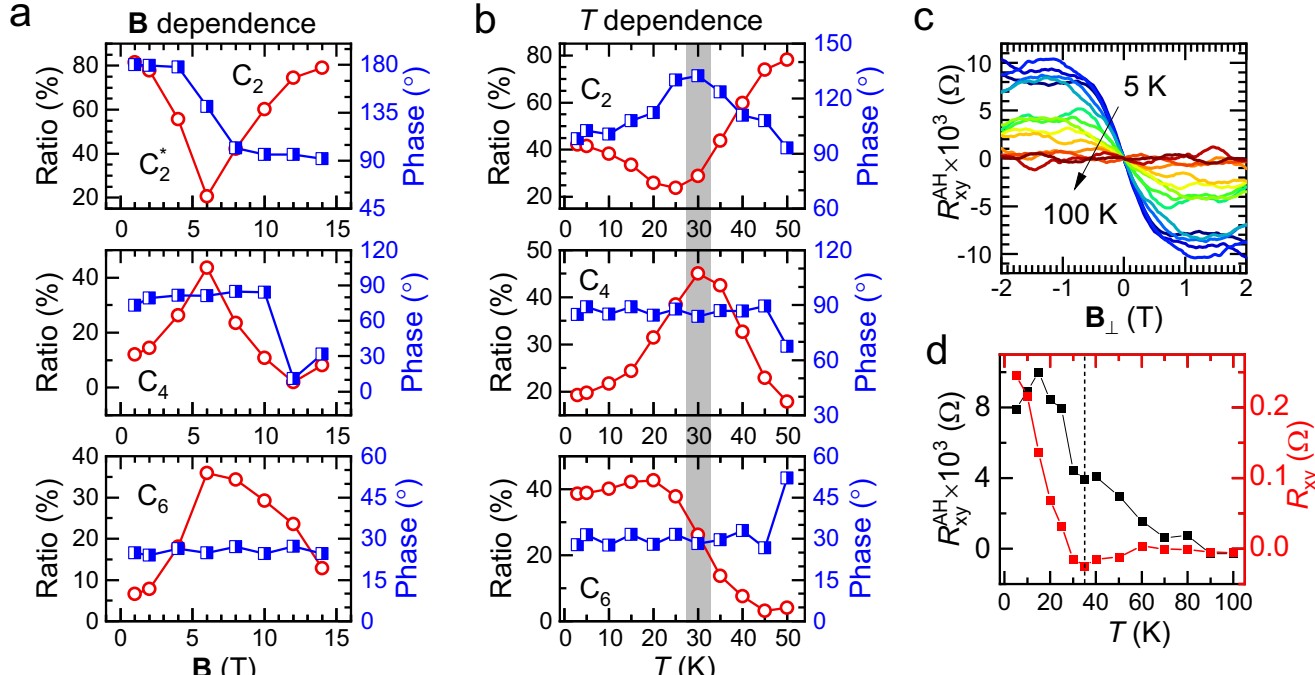

**Fig. 2 | Symmetry components of the in-plane MR vs. B and T. a** The normalized proportions ($\xi_i/(\xi_1 + \xi_2 + \xi_3)$, red open circles) and their phases ($\eta_1$, $\eta_2$ and $\eta_3$, blue squares) of the C2, C4 and C6 symmetry components of the in-plane MR vs. **B** at $T = 5$ K, respectively. Here $i = 1, 2, 3$ represents the upper (C2), middle (C4), and lower (C6) panels. **b** The normalized proportion (red open circles) and its phase (blue squares) of the C2, C4, and C6 symmetry components of the in-plane MR vs. temperature $T$ at **B** = 9 T. **c** The anomalous Hall resistance vs. perpendicular magnetic field **B**$_\perp$ for a number of temperatures ranging from 5 K to 100 K. **d** The Hall resistance (red dots) and the saturated anomalous Hall resistance (black dots) vs. $T$ at **B**$_\perp$ = 1 T. The blue broken line shows the sharp upturn of the Hall resistance and the anomalous Hall resistance at around 35 K.

respect to the $y$-axis. All measured devices are on the order of ten micrometers in size, less than the typical domain sizes on the order of hundreds of micrometers in the material[27]. This can prevent the contribution of domain wall scattering to MR and facilitate probing the possible intrinsic symmetry breaking through angular MR measurements. Figure 1b shows the temperature-dependent resistance ($R$-$T$) curve of a typical 32-nm-thick CVS device at zero magnetic field and at temperatures ranging from 2 K to 300 K. The curve shows a kink at around 85 K, which corresponds to the CDW transition. The residual resistivity ratio (RRR) reaches 56, and a sharp superconducting transition at $T = 4.3$ K is visible in the inset of Fig. 1b, indicating the high quality of the sample. Compared with bulk samples, the slightly higher $T_c$ and lower $T_{CDW}$ is caused by the weaker interlayer coupling in thin flake samples as evident from Raman measurements (see Supplementary note 9), consistent with previous reports[32].

Figure 1c illustrates the in-plane MR as a function of magnetic field angle ($\gamma$) for various magnetic field strengths (**B** = 1, 2, 4, 6, 8, 12 T) at a temperature of 5 K. This temperature was chosen to avoid the sample entering the superconducting state at low magnetic fields. The MR patterns can be divided into three categories based on the magnetic field strength: low, intermediate, and high. In the low field regime (**B** ≤ 2 T), the in-plane MR is anisotropic and displays a two-fold rotational symmetry (denoted as C2*) with maxima along the $\gamma = 0°$ direction. This C2* pattern is always perpendicular to the current direction in a rounded CVS device with radially aligned electrodes (as shown in Supplementary Note 5). Thus, it is confirmed to be a classical MR arising from the Lorentz force effect and related scattering enhancement of charge carriers in CVS under magnetic fields[33,34] (details in Supplementary note 3 and 5). In the intermediate field regime (2 T ≤ **B** ≤ 8 T), the in-plane MR symmetry changes dramatically and complex patterns emerge with increasing **B**. In the strong field regime (**B** ≥ 8 T), the in-plane MR exhibits a strong two-fold anisotropy with maxima

along the $\gamma = 90°$ direction (denoted as C2 to differentiate from the C2* pattern at the low field regime) and a six-fold rotational symmetry with maxima along $\gamma = 30°$, 90° and 150° directions (denoted as C6). Note that the above-mentioned symmetries of the in-plane MR reflect the respective symmetry of various orders in $CsV_3Sb_5$ (see Supplementary note 11), and can be characterized as various angular-dependent components of the in-plane MR[35,36]. We also note that higher-harmonics of the C2 term with small amplitudes cannot be completely ruled out, which might exhibit as C2n terms, where $n = 2, 3, 4, 5...$, and these higher-harmonic terms should behave similarly to the C2 term when subject to changes in external magnetic field and temperature due to the fact that they have the same physical origin.

Figure 1d plots the in-plane MR as a function of the magnetic field angle for a 9 T rotating magnetic field at various temperatures ($T = 5, 15, 25, 35, 45$, and 50 K). The first panel on the left in Fig. 1d shows in-plane MR similar to the MR in the high field regime in Fig. 1c, with strong C2 and C6 components. As the temperature increases from 5 K, the C6 component of the in-plane MR diminishes while the C2 component remains strong. Interestingly, this reduction in the C6 component occurs at around 35 K, which is believed to be associated with electronic nematic order transition[8,37] or time-reversal symmetry-breaking charge order transition[10,24]. On the other hand, we found that the C2 component at high fields persists at higher temperatures and is associated with the symmetry-broken CDW order, as it diminishes together with the CDW order at $T > 85$ K (as shown in Supplementary Fig. 13).

To examine the changes in the in-plane MR symmetry in CVS crystals, we use the following equation to fit to the in-plane MR data:

$$MR = \alpha + \xi_1 \cos\{(2(\gamma + \eta_1))\} + \xi_2 \cos\{(4(\gamma + \eta_2))\} + \xi_3 \cos\{6(\gamma + \eta_3)\} \quad (1)$$

Here $\alpha$ is an offset constant, $\xi_1$, $\xi_2$ and $\xi_3$ represent the magnitude of two-fold (C2 or C2*), four-fold (C4), and six-fold (C6) symmetric in-

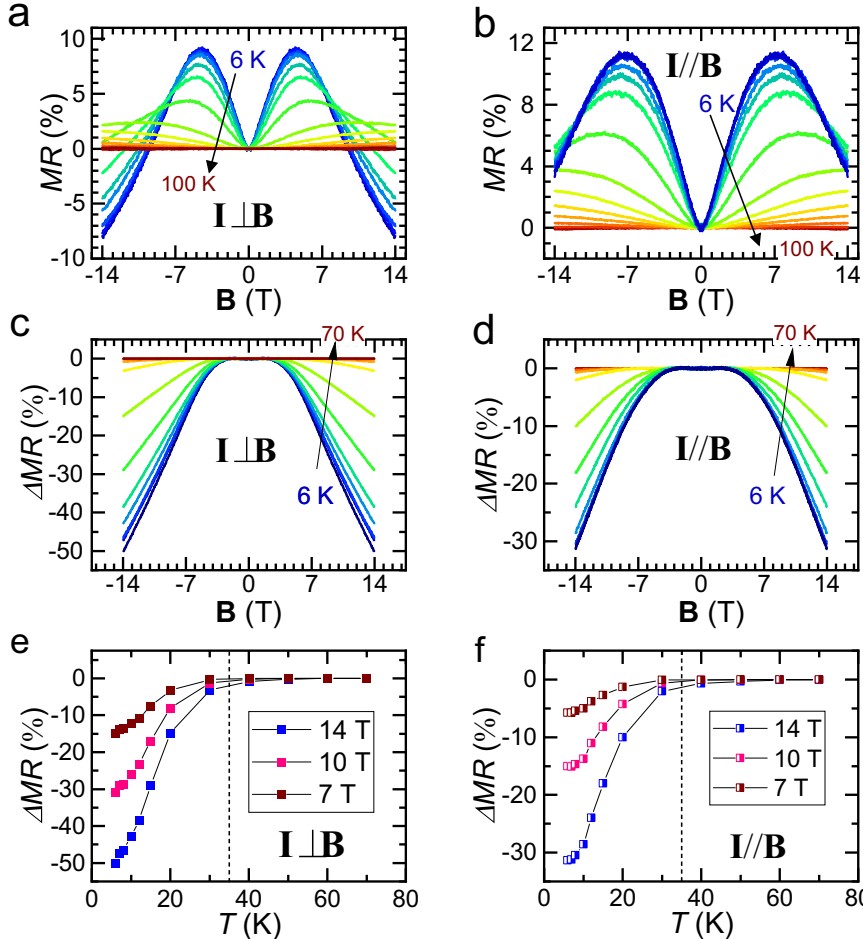

**Fig. 3 | The in-plane negative MR of CVS. a**, **b** In-plane MR at various $T$ with $B$ perpendicular to the current direction ($\gamma = 0$; Fig. 3a) and parallel to the current direction ($\gamma = 90°$; Fig. 3b), respectively. **c**, **d** The net negative in-plane magnetoresistance by subtracting the fitted positive component, i.e., $\Delta MR = MR(\mathbf{B}) - MR_{\text{fitted}}(\mathbf{B})$. **e**, **f** $T$ dependent $\Delta MR$ for $\mathbf{B} = 7, 10, 14$ T. The $\mathbf{B}$ field is perpendicular (Fig. 3e) and parallel (Fig. 3f) to the current direction, respectively.

plane MR components, respectively. The angle $\gamma$ is defined as described in Fig. 1a. $\eta_1$, $\eta_2$ and $\eta_3$ are phases describing relative rotations of respective symmetry components from $\gamma = 0$ (along the *y-axis*). The fitted curves are shown together with the experimental in-plane MR in Supplementary Fig. 6 and the fitted parameters allow us to determine the magnetic field and temperature dependence of the magnitudes and phases of the C2(C2*), C4, and C6 components in the in-plane MR, which provide valuable insights into the electronic symmetry transitions in CVS.

Figure 2a shows the magnetic field dependence of the relative strength of C2, C4, and C6 in-plane MR components at $T = 5$ K. The first remarkable feature is the suppression of the C2* component at $\mathbf{B} = 6$ T (Fig. 2a, upper panel), accompanied by a 90° shift in its phase. As discussed in Fig. 1c, this shift suggests that the low-field C2* component (arising from the classical Lorentz force effect) and the high-field C2 component (associated with the CDW phase) have different origins and are orientated 90° apart in this particular device. The 90° rotation is coincidental and can be altered by changing the current direction (as shown in Supplementary Note 5 and Supplementary Fig. 13). As the magnetic field increases, the C2 component rapidly increases and takes over the proportional weight of the C2* component, causing a dip in its magnitude and a jump in its phase signal during the transition. This trend is even clearer when analyzed in conjunction with the C4 data in the middle panel of Fig. 2a. The C4 component has a peak at $\mathbf{B} = 6$ T, revealing that it is the result of competition between the C2* and C2 components. At low and high

fields, the C4 component is suppressed since either the C2* or C2 component dominates.

The lower panel of Fig. 2a shows the relative strength of the C6 component vs. **B**, which first increases and then decreases with increasing magnetic field, with a peak at $\mathbf{B} = 6$ T. The phase $\eta_3$ of the C6 component remains unchanged over the range of 1 T to 14 T, indicating the stability of this C6 symmetry order. This C6 component involves three potential origins: 1) the symmetry of the kagome lattice, 2) three C2 order domains with angles of 120° between them, and 3) the symmetry of a possible orbital current order. The first mechanism can be easily excluded since the kagome lattice structure is unlikely to be altered by an in-plane magnetic field of ~ 10 T. For the second scenario, it is very difficult for us to directly examine the existence of domains. However, the electron scattering rate by each domain per unit area should be uniform and they contribute the MR of the devices. Then assuming the second scenario is correct, the magnetic field and temperature dependence of C2 and C6 ought to exhibit comparable behavior (due to the fact that they are all from C2 domains), which is inconsistent with our experimental results. Therefore, we believe that the C6 component of MR does not originate from C2 domains, or, at least, the dominant C6 signal is not from C2 domains. In the context of current research, orbital current order could stand as the most plausible candidate mechanism for the C6 component of the MR. The decrease of the C6 component may be understood as follows: Orbital current order can be intuitively regarded as the spontaneously circular motion of electrons within the kagome lattice. When the applied in-

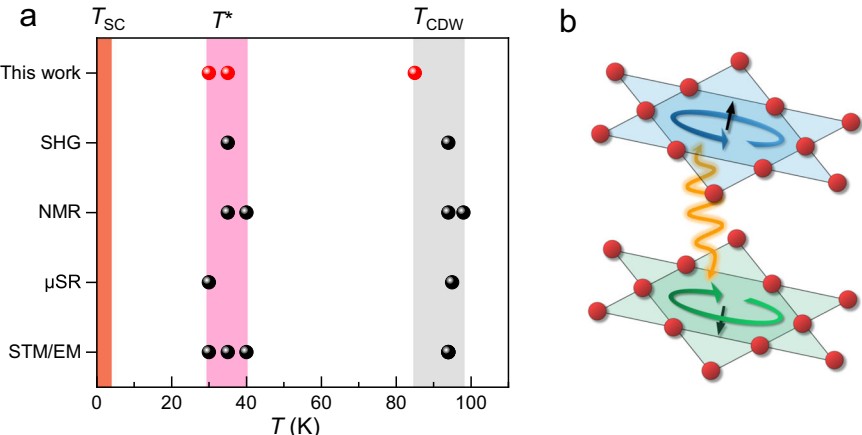

**Fig. 4 | The phase transitions in CVS. a** This graph summarized the phase transitions in CVS reported in the literature (black dots) and in this work (red dots). Source of data from the literatures: STM/EM (Elastoresistance measurements): ref.[3,8], µSR: ref. 24, NMR: ref.[8,47] and SHG: ref.[10]. **b** Schematic diagram of the orbital current order in CVS with three-dimensional coupling.

plane magnetic field is strong enough, electrons tend to move towards out-of-plane and this in-plane circular motion may be disturbed, leading to a decrease of the relative strength of the C6 component as compared to the C2 component from CDW (robust under magnetic field).

Figure 2b depicts the temperature dependence of the C2, C4, and C6 in-plane MR components at $B = 9$ T. A noticeable observation is the sudden change in both the relative strength and phase of the C2 component at around 30 K, which is consistent with the recent report of an electronic nematic order transition[8]. Above the nematic transition temperature ($T_{nem}$), the C2 signal originates from the three-dimensional CDW state that has the same in-plane modulation but with a $\pi$-phase shift between two neighboring CDW planes.

The C6 component, on the other hand, has the steepest rise at ~ 30 K and its magnitude reaches saturation at low temperatures while its phase remains largely unchanged. Previous µSR and optical experiments[7,12,24,28] have confirmed the existence of time-reversal symmetry (TRS) breaking orders in $AV_3Sb_5$, which the anomalous Hall effect in these materials is partly attributed to. We also performed Hall effect measurements at various temperatures to investigate TRS breaking orders in the CVS samples (as shown in Supplementary Fig. 9). Anomalous Hall (or Hall anomaly, HA) resistance $R_{xy}^{AH}$ was obtained by subtracting the linear Hall component in a perpendicular magnetic field $\mathbf{B}_\perp$ ranging from $-2$ T to 2 T (Fig. 2c). As can be seen in Fig. 2c, a finite $R_{xy}^{AH}$ is observed at low temperature and it diminishes at higher temperatures (e.g., $T > \sim 90$ K). Furthermore, $R_{xy}^{AH}$ saturates for $\mathbf{B}_\perp$ higher than 1 T. Figure 2d plots the temperature dependence of both $R_{xy}^{AH}$ and Hall resistance $R_{xy}$ at $\mathbf{B}_\perp = 1$ T. As shown in Fig. 2d, the AHE or HA in the CVS sample emerges when the temperature drops below $T_{CDW}$, consistent with previous reports[23]. Interestingly, both $R_{xy}^{AH}$ and $R_{xy}$ exhibit a sudden increase for $T < 35$ K, which coincides with the emergence of the C6 component of the in-plane MR. This suggests that the C6 component of the in-plane MR, the Hall resistance, and anomalous Hall resistance are all responsive to the emergence of the hidden phase at and below ~ 35 K. The subtle signal of the C6 component above 35 K may be attributed to thermal fluctuations of the hidden order. In addition, we noticed that the concurrent transitions of C2 and C6 components observed near 35 K can be explained by the out-of-phase combination of bond charge order and orbital current order, as proposed in a more recent STM study[38].

To better understand the hidden phase transition in CVS, we have plotted the in-plane MR at various temperatures with the magnetic field perpendicular to the current direction ($\gamma = 0°$; Fig. 3a) and parallel to the current direction ($\gamma = 90°$; Fig. 3b). The in-plane MR increases

with increasing magnetic field for $B < \sim 6$ T, then a remarkable peak in MR appears at around 6 T, and the differential MR becomes negative in the high magnetic field regime. As established from our analysis of the in-plane MR symmetry, classical MR, caused by the Lorentz force, dominates the magneto-transport behavior[39] for $B < \sim 6$ T (more discussions in Supplementary Notes 3 and 5). On the other hand, unconventional MR, originating either from the nontrivial topology, the hidden phase, or the CDW phase, becomes more prominent than the classical effect of the Lorentz force for $B > \sim 6$ T.

We first look into the contribution of the Lorentz force effect. Such effect usually manifests itself in changing the electron trajectory and effectively decreasing the electron mean free path, resulting in quadratic magnetoresistance[39]. For thin film materials under in-plane magnetic field, increased carrier scattering from the top and bottom thin film surface could be caused by the diffusive charge carriers acquiring additional velocity perpendicular to the sample plane[33]. In materials with small Fermi surface or Fermi surfaces with sharp corners[40] and in CDW materials[41–43], linear MR is frequently found.

Therefore, we use the empirical formula $MR = A\left[\sqrt{\left(\mathbf{B}^2 + m^2\right)} - m\right]$ to fit to the in-plane MR in the small field regime, where $A$ and $m$ are fitting parameters, $\mathbf{B}$ is the magnetic field. We found that the in-plane MR in the small magnetic field regime is well described by the above formula (see Supplementary Note 3 for more details).

With the above information, the unconventional contribution to the magnetoresistance (MR) in CVS was determined by subtracting the MR caused by the Lorentz force from the experimental MR data, as shown in Fig. 3c, d. The resulting negative MR was up to $-50\%$ and showed two distinct features: 1) it was observed in both $I\perp\mathbf{B}$ and $I//\mathbf{B}$ configurations; 2) it did not follow the $-\mathbf{B}^2$ form in either the raw data (Fig. 3a, b) or the net negative MR data (Fig. 3c, d). These observations suggest that the negative MR did not arise from a topological effect such as chiral anomaly in Weyl semimetals[44–46]. The change in negative MR, $\Delta MR = MR\,(\mathbf{B}) - MR_{fitted}\,(\mathbf{B})$, was plotted in Fig. 3e, f for magnetic fields of 7, 10, and 14 T, for $I\perp\mathbf{B}$ and $I//\mathbf{B}$ configurations, respectively. Regardless of the magnitude and direction of the magnetic field, negative $\Delta MR$ emerged and rapidly increased below ~ 35 K, in agreement with the temperature of the hidden phase transition. These results suggest that the negative MR is closely related to the hidden phase at and below ~ 35 K.

As previously mentioned, recent experiments have suggested the presence of new phase transitions between the superconductivity and CDW transitions in CVS. We plot our experimental findings together

with previous experimental data in Fig. 4a. The red dots in Fig. 4a are from our transport data and the black dots are from previous reports by other experimental techniques[3,8,10,24,47]. There exists a significant phase transition at $T^* \sim 30\text{-}40$ K, which exhibits complex temperature and magnetic field dependence of in-plane MR with both C2 and C6 rotational symmetric components, as well as a peculiar negative in-plane MR for $T < \sim 35$ K. The C2 component is present at and below $\sim 85$ K (Supplementary Fig. 13 & Fig. 1d) and is linked to the CDW phase, while the C6 component appears only below $\sim 35$ K (Fig. 2b, lower panel). Previous STM, NMR, and EM studies[8] suggest that the phase transition at $T^*$ is due to electronic nematic ordering, while previous µSR experiments point to the formation of a hidden flux phase[7,24]. However, electronic nematic order is not known to cause a negative MR, and the magnetic moments in orbital current order is pointed out-of-plane, and is generally considered not responsive to in-plane magnetic fields.

The apparent contradiction in the observations can be reconciled by taking into account the strong interlayer interactions present in CVS[5,48,49]. These interactions result in a 3D Fermi surface[49] (also see Supplementary note 2) and a 3D CDW[5,48], and drive the fluctuations of the orbital currents[16,17] which lowers the energy of the layered system, much like van der Waals interactions lower the energy of van der Waals crystals[50]. The fluctuating chiral flux in the CVS layers scatters the charge carriers, and an in-plane magnetic field reduces interlayer transport mean free path, suppressing the flux fluctuations, leading to weaker scattering and negative in-plane magnetoresistance, as depicted in Fig. 4b. Recent experimental and theoretical studies have shown that an orbital current order with a negligible but non-zero out-of-plane magnetic field can produce non-zero orbital magnetization in the in-plane direction[10], which strongly couples to in-plane magnetic fields. Another important evidence for the 3D nature of the hidden phase comes from the thickness dependence of the anisotropic in-plane MR. Previous studies have shown that the dimensional crossover from 3D to 2D occurs at thickness $\leq 30$ nm in CVS[31]. Thus, we investigated the in-plane MR in the thinner sample with a thickness of $\sim 20$ nm. In contrast to the 3D samples with a thickness greater than 30 nm, this thinner sample has no C2 component, C6 component or linear negative MR (see Supplementary note 10), pointing to the fact that this hidden order cannot exist in 2D samples.

## Summary

In summary, our study uncovered the presence of anisotropic in-plane magnetoresistance in thin CsV₃Sb₅ crystals, which displays temperature and magnetic field dependence with rotational symmetrical components. The C2 component appears concurrently with the CDW order and shares the same symmetry as the electronic nematic order[8]; Meanwhile, the C6 component emerges at temperatures below $\sim 35$ K[7,16,24] and reflects the orbital current order's spatial symmetry. Below this temperature, a quasi-linear, non-saturating negative in-plane magnetoresistance also manifests in both $\mathbf{I} \perp \mathbf{B}$ and $\mathbf{I}//\mathbf{B}$ configurations, indicating a three-dimensionally interacting, magnetic field-tunable orbital current ordered phase. Our findings, combined with prior works using various experimental techniques[8,10,24,47], provide a complete profile of the physical properties of this hidden phase in CVS.

## Methods

Device Fabrication: Al₂O₃-assisted exfoliation techniques were used to obtain thin flakes of CVS crystals. Firstly, the Al₂O₃ film was deposited by thermal evaporation onto a freshly prepared surface of the bulk crystals and then a thermal release tape was used to pick up the Al₂O₃ film, along with pieces of CVS microcrystals separated from the bulk. The Al₂O₃/CVS stack was subsequently released onto a piece of transparent polydimethylsiloxane (PDMS) film. Finally, the PDMS/CVS /Al₂O₃ assembly was stamped onto a substrate and the PDMS film was quickly peeled away, leaving the Al₂O₃ film covered with freshly cleaved CVS flakes on the Si/SiO₂ substrate. To accurately measure electrical

transport properties, we used a tungsten needle to cut the flakes into long strip shapes for Hall bar devices and used an AFM tip to cut the flakes into the solar shapes for circular disk devices. The measured sample thicknesses are within the range of 10 nm to 100 nm. The typical device images were shown in Supplementary Figs. 8a and 14a. Then standard e-beam lithography was used to pattern electrodes, followed by e-beam evaporation of Ti (5 nm) and Au (100 nm). The device fabrication process was carried out in an inert atmosphere and vacuum to minimize sample oxidation, and samples were briefly exposed to air only under the protection of a PMMA capping layer.

Magnetoresistance Measurement: Transport measurements were conducted at temperatures between 1.5 K and 300 K with magnetic fields up to 14 T using an Oxford Teslatron cryostat and a Quantum Design PPMS. Lock-in amplifiers were used to measure longitudinal resistance ($R_{xx}$) and Hall resistance ($R_{xy}$) at a frequency of 77.77 Hz. Changing the magnetic field direction was achieved by rotating the sample holder. The thickness of the various samples was measured using Atomic Force Microscopy (AFM).

## Data availability

Data for figures that support the current study are available at https://doi.org/10.7910/DVN/RB9B1G.

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

## Acknowledgements

This project has been supported by the National Key R&D Program of China (Grant No. 2019YFA0308402), the Innovation Program for Quantum Science and Technology (Grant No. 2021ZD0302403), the National Natural Science Foundation of China (NSFC Grant Nos. 12304055, 11934001, 92265106, 11921005). J.-H.C. acknowledges technical support from Peking Nanofab.

## Author contributions

J.-H.C. conceived the idea and directed the experiment. Z.W. and Y. Y. provided high-quality crystals. X.W. and C.T. fabricated most of the devices and performed the transport measurements. H.C., Y.K.L., S.L., Y.S., Y.F., and M.H. aided in sample preparation and transport measurement. Y.Z. & Q.X. added in Raman measurement. Y.L., Y.Y., and X.C.X. provided theoretical analysis. J.-H.C., X.W., C.T., and H.C. collected and analyzed the data. X.W. and J.-H.C. wrote the manuscript. All authors commented and modified the manuscript.

## Competing interests

The authors declare no competing interests.
