## [Peer Review File · Nature Communications]

REVIEWER COMMENTS

Reviewer #1 (Remarks to the Author):

In this work, X. Wei et al. study the magnetotransport of CsV₃Sb₅, and report the signal of a hidden phase below 35 K, which is characterized by the observation of six-fold rotational symmetry of in-plane magnetoresistance and in-plane negative magnetoresistance. This phase is proposed to be related to the orbital current order. The results in this work are intriguing. I can recommend publication if the questions below are addressed:

1. The angle dependent magneto-transport of CsV₃Sb₅ has been studied in Nat. Commun 12, 6727 (2021), which has reported C₂ symmetry and signal of C₆ symmetry with current applied along c axis. The authors should cite this highly related work in the manuscript. In addition, in that work, the six-fold oscillation of angle dependent magnetoresistance was proposed to be related to the C₆ symmetry of the crystal structure. In this manuscript, the extracted C₆ component persists above 35 K (Fig. 2b), this seems conflict with the claimed new phase with six-fold rotational symmetry below 35 K. The authors should elucidate why C₆ component appears at high temperature and whether the observed it is from six-fold crystal structure or Fermi surface.
2. It's not clearly discussed why the C₂ and C₆ symmetry appear at high magnetic field. Also, in Fig.2a, the ratio of all the extracted symmetry components change around 6 T, is 6 T a special point that influence the order states? And why does the ratio of C₆ component decrease at higher field? These questions and the possible origin of field dependence of different order states should be discussed.
3. TRSB was widely reported to appear just below the CDW transition, such as in Ref. 7 and Phys. Rev. Research 4, 023244 (2022). But in this manuscript, the phase transition around 35 K is claimed to be related to the time-reversal symmetry breaking (TRSB). The TRSB should be carefully discussed.
4. In the circular disc devices, the C₂ symmetry doesn't change with current directions. This should be related to the order state along specific direction. What's the relation between this C₂ symmetry and crystalline axis in kagome lattice of CsV₃Sb₅?
5. The negative MR appears when applying in-plane magnetic field. I suggest to analyze the angle dependence of MR when rotating magnetic field to z direction, to see how negative MR changes with angle, and whether the angle dependent MR is consistent with the proposed theory of fluctuations of the orbital currents.
6. Does the negative MR have relation with the applied current direction in CsV₃Sb₅?

Reviewer #2 (Remarks to the Author):

In their paper «Three-dimensional hidden phase probed by in-plane magnetotransport in kagome metal CsV₃Sb₅», X. Wei et al. investigate the Kagome metal CsV₃Sb₅ (CVS) using in-plane magnetotransport. By varying the field angle, the symmetry of the conductor is deduced and a phase transition around 35K is claimed based on the symmetry of the AMR patterns. CVS, and in particular the transition around 35K is an extremely actively debated topic, hence the paper is highly topical. However, I do not find the main conclusions sufficiently reflected in the data.

Major concerns:

- Significant strain contaminates the data. It is clear that the device is under significant in-plane strain from thermal mismatch to the substrate. The T_c and T_{CDW} of CVS is well established in bulk crystals and is consistently observed by multiple groups and growth methods. Their values are 2.5K and 94K, respectively (e.g. B. Ortiz et al, PRL 124, 247002 (2020)). In these structures, 4.3K and 85K are observed. Contrary to the authors statement, these values do not “indicate the high quality of the sample” but evidence massive amounts of strain. This means in the sample the CDW is significantly suppressed, which is known to favor superconductivity. Such a large shift corresponds to ~ 0.5 GPa of hydrostatic pressure (Nat. Comm 12:3645 (2021)) or about 1% uniaxial compression (PRB 104, 144506 (2021)). CVS and its orderings are known to be extremely sensitive to external perturbations, and hence the physics under these conditions here does not resemble that of the bulk state at all. Reliable quantification of this is very difficult in flakes, and even if possible, I see no convincing way to make any statements about symmetry breaking through the T' phase given that the rotational symmetries of the Kagome are already substantially broken by strain (assuming that not the highly unlikely case materialized in which the uncontrolled pressure happens to be hydrostatic).

- I cannot follow the symmetry arguments. A symmetry is either present or broken, and in this experiment it is throughout all parameter space explicitly reduced to C₂, at all fields and temperatures. The Lorentz force owing to the authors choice of plane of rotation necessarily causes an asymmetry between transverse and longitudinal that arises purely from kinematics and is independent of the crystal symmetry of the probed solid. Once the symmetry of the data is lowered to C₂, it is logically impossible to deduce a “further” symmetry lowering by fitting another $\cos(\theta)$ at a different phase and claiming a coexistence of multiple C₂ objects. The angle-dependence of MR is a very complex function, especially with complex Fermi surfaces and confined geometries / finite size effects. This complex object naturally can be decomposed into a Fourier series. The higher order terms of such decomposition correspond to structure beyond $\cos(\theta)$, and they fundamentally cannot be interpreted as an “additional” symmetry, such as here argued C₆ or a “second C₂”. The proper symmetry of this dataset necessarily is C₂ due to the chosen plane of rotation.

Minor points:

- Device fabrication. The discussed physics is bulk, and a discussion why these measurements were performed on flakes is lacking. Especially with all the issues from fabrication, it is strange this measurement was not done on macroscopic crystals. Please comment on the reason behind it and bind the fabrication more into the manuscript. The device fabrication and details (geometry?) are missing, Figure S1b lacks any scale information. Without a clear description on the fabrication steps it is very difficult to assess the work.

- Unknown values have been subtracted from all data in the polar plots, making it impossible to understand the magnitude of the effect on the overall resistance. Please plot proper MR as a dimensionless quantity.

- What is the crystallographic alignment of the device? Is the C6 component in registry with the Kagome net or rotated at an arbitrary angle.

In conclusion, I cannot support the publication of the manuscript. It is unclear what exactly one learns from this experiment. The transport situation is extremely complex (finite size, complex Fermi surface, varying Lorentz force effects, substantial strain) so that quite naturally very complex patterns in the angle-dependent MR are to be expected and are observed. Disentangling them into microscopic insights is very difficult. Transport probes everything, hence it is not surprising that these experiments detect anomalies at T_{CDW} , T' and T_c ; in accordance with the well established literature about them. It is difficult for me to pinpoint, setting the methodological issues aside, what new insights this work provides. A broken C2 state at T' has already been seen by multiple groups, e.g. Y. Xiang et al., Nat. Comm. 12:6727 (2021), yet even that is difficult to state conclusively given that in the presented experiment the apparent symmetry is always C2 even in higher symmetric metals such as copper.

Reviewer #1 (Remarks to the Author):

In this work, X. Wei et al. study the magnetotransport of CsV₃Sb₅, and report the signal of a hidden phase below 35 K, which is characterized by the observation of six-fold rotational symmetry of in-plane magnetoresistance and in-plane negative magnetoresistance. This phase is proposed to be related to the orbital current order. The results in this work are intriguing. I can recommend publication if the questions below are addressed:

Reply: We are grateful to the reviewer for his/her positive remarks and recommendation for the publication of our manuscript after revision. We have carefully analyzed and addressed all the comments provided by the reviewer, and made necessary modifications to our manuscript accordingly. We believe that we have successfully addressed all questions and comments raised by the reviewer. The point-by-point reply is listed below.

1. The angle dependent magneto-transport of CsV₃Sb₅ has been studied in Nat. Commun. 12, 6727 (2021), which has reported C₂ symmetry and signal of C₆ symmetry with current applied along *c* axis. The authors should cite this highly related work in the manuscript. In addition, in that work, the six-fold oscillation of angle dependent magnetoresistance was proposed to be related to the C₆ symmetry of the crystal structure. In this manuscript, the extracted C₆ component persists above 35 K (Fig. 2b), this seems conflict with the claimed new phase with six-fold rotational symmetry below 35 K. The authors should elucidate why C₆ component appears at high temperature and whether the observed it is from six-fold crystal structure or Fermi surface.

Reply: We thank the referee for his/her rigorous and careful comments. We have cited this related work (Ref. [26]) as suggested in the revised manuscript.

Overall, Ref. [26] emphasized on vertical transport properties of CsV₃Sb₅ bulk crystals and most attention was focused on the twofold symmetry of resistivity along the *c*-axis in the superconducting state and in the normal state, as well as on the possible origins of such nematicity. There were some discussions on the C₆ component of such vertical resistivity under in-plane magnetic field, which are in a number of ways different from our experiment.

- 1) First of all, our work focused on the in-plane resistivity under in-plane magnetic field, by changing both the current directions and the magnetic field directions to separate various contributions with classical and/or quantum origins. In Ref. [26], since the current was applied along the *c*-axis direction only, the magnetoresistance provides information on the *c*-direction electronic states.
- 2) Second, our work observed peculiar linear negative in-plane MR, and the C₆ contribution we observed is mainly from this negative in-plane MR. On the contrary, such negative in-plane MR

is not observed in the vertical magneto-resistivity of Ref. [26], thus its symmetry could not possibly be revealed by Ref. [26].

- 3) Third, our work focused on the normal state MR while Ref. [26] mostly focused on the superconductivity of the crystals.

With the above differences, together with limit data on the C6 component reported in Ref. [26] (as it is not the main focus of Ref. [26]), it is difficult to tell if the C6 component found in our work can be associated with the vertical resistivity measured in Ref. [26].

Out of the above uncertainty, what is certain is that the C6 component we observed in this work is not from the kagome crystal structure, with the following three strong evidences:

- 1) Non-monotonic magnetic field dependence. The amplitude of the C6 component first increases and then decreases with increasing magnetic field, with a peak at $B = 10$ T (see **Figure S4.2a**). The kagome lattice structure is unlikely to be altered by an in-plane magnetic field of ~ 10 T.
- 2) The temperature dependence. If the observed C6 component arises from the crystal structure, then it should exist throughout all temperatures. However, the C6 component is invisible above 35 K in our experiments (see **Figure S4.2b**).
- 3) Its sample thickness dependence. We have performed further in-plane MR experiment with thinner samples (~ 20 nm) and discovered sharp contrast to thick samples. The thinner sample (~ 20 nm) exhibits no negative magnetoresistance, but rather, a two-fold symmetric positive MR at different magnetic fields (see **Figure R1** below and **Figure S10** of revised supplementary information). Our previous work (2D Mater. 10, 015010 (2022)) has confirmed that a dimensional crossover in the electronic states from 3D to 2D occurs as the sample thickness decreases to ~ 30 nm and below. 2D fluctuations in thinner samples likely destroy the 3D hidden order, leaving only the C2 component to be visible in the data.

All of the above evidence unambiguously shows that the C6 components observed in our work is not related to the crystal structure.

Figure R1. In-plane MR in CsV_3Sb_5 with the thickness of ~ 20 nm. **a**, Polar plot of the in-plane MR at different magnetic field orientations at a number of magnetic field values from 1 T to 14 T.

The current direction is marked in the plot, showing its origin as the Lorentz effect. **b**, The positive, monotonic, and quadratically increasing MR further demonstrates its classical origin.

We agree with the reviewer that there is still a subtle signal of C6 component at 40 K and even at 45 K. But at 50 K and above, the C6 component is completely invisible (see **Figure 2b** and **Figure S4.2b**). We speculate that this subtle signal above 35 K may be attributed to the remanence of the hidden order in a relatively broad transition. We have modified the manuscript according to the reviewer's comments, adding the following sentence at Page 8, Line 201 in the revised manuscript: "The subtle signal of the C6 component above 35 K may be attributed to thermal fluctuations of the hidden order."

2. It's not clearly discussed why the C2 and C6 symmetry appear at high magnetic field. Also, in Fig.2a, the ratio of all the extracted symmetry components change around 6 T, is 6 T a special point that influence the order states? And why does the ratio of C6 component decrease at higher field? These questions and the possible origin of field dependence of different order states should be discussed.

Reply: We thank the reviewer for bringing up this important point. It is indeed highly necessary to have a clear and detailed discussion on the magnetic field dependence of the C2 and C6 components. In the following, we provide a simplified model to explain the unusual symmetries at different magnetic fields. This is mainly a semi-classical model involving a) charge carrier scattering with isotropic impurities and b) band structure modification by CDW and the hidden order. A more elaborate model would be a highly important future theoretical work that we hope this paper could stimulate.

1) C2* component (Lorentz force effect). This classical effect only needs to involve scattering from isotropic impurities. For an isotropic and nonmagnetic material, the electrons in this material are mainly scattered by phonons and static impurities. Classical physics tell us that only electrons with v_x (here we define the x -direction to be the current direction) contribute to the resistance. According to Lorentz force formula $q(v \times B)$, when applying in-plane B perpendicular to current I , electrons with v_x move in circular loops in x - z plane. The scattering probability $1/\tau$ (from phonon and static impurities) of electrons with v_x will increase and consequently, MR with $B \perp I$ will increase with B . However, as $B // I$, B only change the scattering probability of electrons with v_y , and consequently MR with $B // I$ almost doesn't depend on B . Rotating the in-plane magnetic field would result in a C2 symmetric change in the absolute value of B_y (B component along the y -direction), therefore the in-plane MR also exhibits a C2 symmetry. From the above discussion, in-plane MR induced by Lorentz force has two obvious features: The C2 symmetry direction is determined by the current direction; its maxima located at $B \perp I$, and it should appear with or without the CDW transition. In our

work, such contribution is marked as C2* component, and its behavior is absolutely consistent with Lorentz force picture (see supplementary Figure S5).

- 2) C2 component (nematic order from CDW and/or electronic nematicity). Now we consider the situation where the material become anisotropic due to additional ordering. Assuming the material with a C2 symmetry order and the order direction has an angle of α degrees with respect to the current direction, as shown in **Figure R2a**. In **Figure R2a**, we define the current direction to be the x-direction, and y-direction to be perpendicular to the x-direction. It is reasonable to hypothesize that electrons moving perpendicular to the order direction (marked as $v_{x,\perp}$ in **Figure R2a**) have larger effective mass (band modification) and/or stronger scattering rate (combined impurity effects with band modification) than along the parallel direction (marked as $v_{x,\parallel}$). Here v_x denotes the average velocity of the charge carriers under current I . In this case, we only need to consider the effect of magnetic field on $v_{x,\perp}$. Similar to the case in 1), the scattering probability $1/\tau$ from C2 order is mainly related to B component along the C2 direction, inducing a MR with C2 symmetry. In terms of band structure modification, the magnetic field creates helical carrier trajectory which increases the weight of transport along the magnetic field direction, i.e., forcing more electrons along the direction along the magnetic field with an effective mass changing with C2 symmetry, creating similar effects to that of the impurities scattering. Therefore, in-plane MR vs. rotating B angle induced by a C2 symmetric order is also of C2 symmetry. In contrast to C2* MR which is defined by the direction of the current, the C2 component would have a direction determined by the C2 nematic order. **Figure R2b** shows the in-plane MR in two special situations where $\alpha = 90^\circ$ (upper panel) and 0° (lower panel) in CsV₃Sb₅. When $\alpha = 90^\circ$, under a rotating magnetic field with angle γ with respect to the y-direction, maximum MR is achieved with $\gamma = 90^\circ$, while minimum MR is achieved with $\gamma = 0^\circ$. Similar situation can be found for $\alpha = 0^\circ$.

Now we use this model to explain the observed C2 symmetry MR of CsV₃Sb₅. In kagome metal CsV₃Sb₅, the C2 symmetry order has been confirmed to be a π phase shift between neighboring layers below T_{CDW} and an electronic nematic order below 35 K. This is consistent with our results in **Figure 1c** of the main text and Supplementary **Figure S5c**, i.e., the C2 symmetry is not determined by current direction, but by the order direction.

Figure R2. In-plane MR model. **a**, Schematic diagram of a device. There is a C2 symmetric order in the sample, and its direction forms an angle of α degrees with the x-direction. The rotating B is defined by an angle γ with respect to the y-direction. **b**, In-plane MR vs. γ when $\alpha = 90^\circ$ and 0° .

- 3) C4 component (coupling effects). As discussed in the main text, the appearance of the C4 component is coincident with C2* and C2 components orientated 90° away from each other. Consider finite coupling between the effect of isotropic impurity scattering and the anisotropic effect by C2 nematic orders, $\rho_{xx} \propto 1/\tau_{\text{impurities}} \cos(2\gamma) + 1/\tau_{\text{nematic order}} \cos(2\gamma + 90^\circ) + \delta(\text{impurities, nematic order})$, where $1/\tau_{\text{impurities}}$ is from impurities scattering, $1/\tau_{\text{nematic order}}$ is from the effective nematic order scattering including band modification effects, and $\delta(\text{impurities, nematic order})$ denotes their coupling term. The coupling term can be simply understood as:

$$\delta(\text{impurities, nematic order}) \propto \frac{(1/\tau_{\text{impurities}})(1/\tau_{\text{nematic order}})}{(1/\tau_{\text{impurities}}) + (1/\tau_{\text{nematic order}})} \cos(2\gamma) \cos(2\gamma + 90^\circ) \\ \propto \cos(4\gamma).$$

At low and high fields, $1/\tau_{\text{impurities}}$ and $1/\tau_{\text{nematic order}}$ dominate and provide C2* and C2 signals, respectively. At the intermediate field, the coupling term reaches maximum value and shows a C4 symmetric MR, which is in accord with our results in **Figure 2a**.

- 4) C6 component (hidden order). The C6 component can be understood in ways similar to the discussion of the C2 component in 2), which is the key finding in our experiment. If the hidden order has three symmetric axes, it would cause the effective scattering probability vs. B to exhibit a six-fold symmetry, providing the C6 component in the in-plane MR. And we have also excluded the possibility of trivial origins of this C6 component, such as from crystal symmetry.

To summarize, the C2 and C6 components of MR are mainly due to the influence of magnetic field changing the effective electron scattering probability, with contribution from both impurity

scattering and from band structure modification, as compares to only the impurity scattering from the classical $C2^*$ component. Thus, as B increases, the $C2$ and $C6$ components become more prominent. The kink of relative strength of each component at 6T is a manifestation of such changes in the dominant contribution of the in-plane MR. We have put the above analysis in the revised Supplementary Information S11.

Decreasing $C6$ component at high field: In kagome metal AV_3Sb_5 , combined with previous researches and our work, the hidden order possesses two features: time reversal symmetry breaking and six-fold symmetry. Meeting these two conditions, the most likely candidate is orbital current order. This order can be intuitively regarded as spontaneous circular motion of electrons within the kagome lattice. When the applied in-plane magnetic field is strong enough, electrons tend to move towards out of plane and this in-plane circular motion may be disturbed, leading to a decrease of the relative strength of the $C6$ component as compares to the $C2$ component from CDW (robust under magnetic field). We have added this discussion at Page 7, Line 174 in the revised main text.

3. TRSB was widely reported to appear just below the CDW transition, such as in Ref. 7 and Phys. Rev. Research 4, 023244 (2022). But in this manuscript, the phase transition around 35 K is claimed to be related to the time-reversal symmetry breaking (TRSB). The TRSB should be carefully discussed.

Reply: We appreciate the careful review by the reviewer. We agree with the reviewer that there is still a heated debate on when the TRSB appears. Since the discovery of kagome metals AVS, whether the time reversal symmetry is broken has been an extremely actively topic. In the early stages, no obvious evidence of magnetic ordering is found in muon spin spectroscopy (μ SR) and elastic neutron scattering measurements; subsequently, indication of TRSB was found in a series of different experiments, such as anomalous Hall effect, STM, SHG, μ SR, and Kerr effect. They account the TRSB for orbital current order. Nevertheless, the onset temperature of orbital current order is not the same in different experiments, e.g. ~ 35 K in SHG (Nature 611, 461) and ~ 92 K ($=T_{CDW}$) in scanning birefringence microscopy (Nat. Phys. 18, 1470). Different groups yielded different results when conducting the μ SR experiment, that the onset temperatures 50K/70K/94K were reported respectively in Phys. Rev. Research 4, 033145; arXiv:2107.10714; Phys. Rev. Research 4, 023244.

Thus, our experiment served to provide a strong transport evidence of TRSB at ~ 35 K, which would help to clarify the current debate.

4. In the circular disc devices, the $C2$ symmetry doesn't change with current directions. This should be related to the order state along specific direction. What's the relation between this $C2$ symmetry and crystalline axis in kagome lattice of CsV_3Sb_5 ?

Reply: We thank the reviewer for this important question. We shall answer the two aspects of this question in the following.

Origin of the C2 order: In kagome metals, there are two possible origins of the C2 symmetry: a π phase shift between neighboring layers and electronically driven nematicity. These two origins take place at distinct temperatures. The interlayer shift occurs at T_{CDW} , while the electronic nematic transition occurs at ~ 35 K (Nature, 604, 59). In our work, we observed the two C2-symmetric transitions as shown in **Figure S8.3** and **Figure 2b**. Significantly, both of these C2 symmetries have the same phase, indicating that the direction of nematicity aligns with the direction of the interlayer shift, which is consistent with the results of STM.

Order direction vs. crystal direction: For a micro-flake sample of CsV_3Sb_5 , it is extremely difficult to accurately measure the relationship between the C2 symmetry and the crystalline axis in the kagome lattice except using scanning tunneling microscopy (STM). Nonetheless, we attempted to determine their directions using Raman measurements. **Figure R3c** (also Supplementary **Figure S9e**) shows the angle dependence of the E_{2g} mode intensities, which reach their maximum value at $\sim 45^\circ$, corresponding to the crystalline axis (Phys. Rev. Research 4, 023215). We plotted the kagome lattice and the in-plane MR in **Figure R3d** (or **Figure S9f**), where the lattice orientation was determined by Raman spectra in **Figure R3c** (or **Figure S9e**). It can be seen that the C2 component of the in-plane MR almost aligns with the crystalline axis of the kagome lattice. We have put the above results in the revised Supplementary Information S9.

Figure R3. Orientation of the CsV_3Sb_5 flake sample detected by Raman spectra and in-plane MR
a, The optical picture of flake sample with Hall bar electrodes. **b**, Raman scattering intensity

measured in parallel polarization with polarization rotated 180° on flake sample. A_{1g} and E_{2g} phonon frequency were observed. **c.** Polarization angle dependence of E_{2g} phonon intensities in flake sample. **d.** In-plane MR vs. kagome lattice. The orientation of kagome lattice is determined by Raman data in **c.** All the Data were collected at 5 K.

5. The negative MR appears when applying in-plane magnetic field. I suggest to analyze the angle dependence of MR when rotating magnetic field to z direction, to see how negative MR changes with angle, and whether the angle dependent MR is consistent with the proposed theory of fluctuations of the orbital currents.

Reply: In the Supplementary Information S2, we have shown the MR with magnetic field rotating from the out-of-plane direction to the in-plane direction. However, the interval of variation in the tilting angle is 10° , which did not clearly show how the negative MR evolves with magnetic field angle tilting out of plane. According to review's suggestion, we have performed additional measurement of MR with small intervals of tilting magnetic field angle Δ from the x-y plane to the z axis. The MR vs. angle Δ is plotted in **Figure R4**. We found that with increasing magnetic field tilted from the x-y plane, the amplitude of the negative MR significantly decreases. Once the tilting angle is above 5° , the negative MR completely vanishes. This phenomenon can be understood as increasing out-of-plane magnetic field strengthens in-plane orbital current order and thereby suppresses its fluctuations, which is also in accord with the proposed theory of fluctuations of the orbital current order. We have put the above results and discussions in the revised Supplementary Information S12.

Figure R4. MR with different angles (Δ) deviation from the x-y plane. When $\Delta = 0^\circ$, the magnetic field is perpendicular with the current direction.

6. Does the negative MR have relation with the applied current direction in CsV3Sb5?

Reply: To study the relation between negative MR and applied current direction, we measured the MR in $B \perp I$ and $B // I$ configurations with different current directions in the circular disc devices (see Supplementary Figure S5a). As for $B \perp I$, negative MR is hardly affected by the direction of the current in **Figure R5b**. Interestingly, negative MR with $B // I$ configuration changes significantly on the direction of the current in **Figure R5a**. Generally, MR from conventional electron-impurity scattering in the $B // I$ configuration is much smaller than that in the $B \perp I$ configuration. Our data suggested that the negative MR must results from an unconventional scattering, which we link to the hidden order in our work. We have put the above results in the revised Supplementary Information 13.

Figure R5. Negative MR with various current directions. **a**, MR with $I // B$; **b**, MR with $I \perp B$.

Reviewer #2 (Remarks to the Author):

In their paper «Three-dimensional hidden phase probed by in-plane magnetotransport in kagome metal CsV₃Sb₅», X. Wei et al. investigate the kagome metal CsV₃Sb₅ (CVS) using in-plane magnetotransport. By varying the field angle, the symmetry of the conductor is deduced and a phase transition around 35K is claimed based on the symmetry of the AMR patterns. CVS, and in particular the transition around 35K is an extremely actively debated topic, hence the paper is highly topical. However, I do not find the main conclusions sufficiently reflected in the data

Reply: We appreciate the reviewer's effort to evaluate our work and are grateful for the constructive comments. According to the reviewer's suggestions and comments, we have performed substantial additional experiments and analysis, and we believe that these changes have significantly improved our manuscript.

Major concerns:

Significant strain contaminates the data. It is clear that the device is under significant in-plane strain from thermal mismatch to the substrate. The T_c and T_{CDW} of CVS is well established in bulk crystals and is consistently observed by multiple groups and growth methods. Their values are 2.5K and 94K, respectively (e.g. B. Ortiz et al, PRL 24, 247002 (2020)). In these structures, 4.3K and 85K are observed. Contrary to the authors statement, these values do not “indicate the high quality of the sample” but evidence massive amounts of strain. This means in the sample the CDW is significantly suppressed, which is known to favor superconductivity. Such a large shift corresponds to ~0.5GPa of hydrostatic pressure (Nat. Comm 12:3645 (2021)) or about 1% uniaxial compression (PRB 104, 144506 (2021)). CVS and its orderings are known to be extremely sensitive to external perturbations, and hence the physics under these conditions here does not resemble that of the bulk state at all. Reliable quantification of this is very difficult in flakes, and even if possible, I see no convincing way to make any statements about symmetry breaking through the T' phase given that the rotational symmetries of the kagome are already substantially broken by strain (assuming that not the highly unlikely case materialized in which the uncontrolled pressure happens to be hydrostatic).

Reply: we appreciate the reviewer's constructive comments regarding the possible presence of strain in our samples and its impact on the experimental results. Indeed, we agree with the reviewer that if strain does exist, our conclusions become not intrinsic to the bulk CsV₃Sb₅. We can show that the measured samples in our main text is NOT affected by strain effect from the following three points:

1. Raman experiment shows no strain effects: To investigate the possible strain effect, we first performed Raman scattering measurements on both flake and bulk samples of CsV₃Sb₅ at 5 K. As shown in **Figure R6b**, two prominent peaks at ~119 cm⁻¹ and 137 cm⁻¹ were detected in both

samples, which correspond to the phonon E_{2g} and A_{1g} modes, respectively, consistent with previous Raman studies (Phys. Rev. Research 4, 023215; Nat. Commun. 14, 2492). The E_{2g} and A_{1g} modes represent the in-plane vibration and the out-of-plane vibration of Sb atoms, respectively. By fitting the data with Lorentz functions, we obtained their precise peak frequency values at different rotation angles of the linearly polarized incident light, as shown in **Figure R6d**. The important finding is that there is no distinguishable difference in the E_{2g} (in-plane vibration) frequency between the flake and bulk samples. This is also clearly shown in the two vertical straight lines in **Figure R6c**. It shows that both the flake and bulk samples have almost identical in-plane crystal structure and thereby the strain effect from the substrate is negligible and cannot reach 1% uniaxial compression as mentioned in PRB 104, 144506.

Figure R6. Raman spectra of CsV_3Sb_5 flake and bulk samples. **a**, The optical picture of flake sample with Hall bar electrodes. **b**, Comparison of Raman spectra obtained in $A_{1g}+E_{2g}$ symmetry for bulk and flake samples. **c**, Raman scattering intensity measured in parallel polarization with polarization rotated 180° on flake sample. **d**, A_{1g} and E_{2g} phonon frequency as a function of polarization angle for bulk and flake samples. All the Data were collected at 5 K.

2. Thinner samples do not show strain effect: Given that the substrate strain will be gradually released as the thickness of the samples increases (J. Appl. Phys. 128, 045303; PRB 104, 024509), thinner samples will experience stronger strain effect. Thus, we measured the in-plane MR in a thinner sample with thickness of ~ 20 nm. The data was plotted in **Figure R7**. Different from thicker flake samples (≥ 30 nm), we only observed classical Lorentz force MR in the 20nm sample (termed C2* in our manuscript), e.g., the MR is always positive, the magnetic field dependence is quadratic, and the minimum MR is along the I//B configuration (see **Figure R7a**). That means, in the thinner sample, 1) we did not observe any strain effect in the MR data, and 2) the dimensional crossover (2D Mater. 10, 015010) destroys the hidden phase, rendering the

C2 and C6 components to vanish together with the negative MR. This fully demonstrates that the unusual anisotropic in-plane negative MR observed in our work does not result from the substrate strain effect.

Figure R7. In-plane MR in CsV_3Sb_5 with the thickness of ~ 20 nm. **a**, Polar plot of the in-plane MR at different magnetic field orientations at a number of magnetic field values from 1 T to 14 T. The current direction is marked in the plot, showing its origin as the Lorentz effect. **b**, The positive, monotonic, and quadratically increasing MR further demonstrates its classical origin.

3. Origin of variation in T_c and T_{CDW} : The differences concerning T_c and T_{CDW} between thin flakes and bulk crystals have been carefully studied in the literatures (PRL 127, 237001; Nat. Commun. 14, 2492), and two different explanations were proposed.

- a) In PRL 127, 237001, the variation of T_c and T_{CDW} is caused by surface oxidation induced hole doping. The doping pushes the van Hove singularity up and the density of state at the Fermi energy increases, leading to enhance T_c . Since Fermi surface nesting becomes weaker, T_{CDW} falls. Since we took extreme care in the device fabrication process, the samples were fabricated in inert atmosphere, and were not exposed to the room atmospheric environment without the protection of a capping layer, we believe the influence of surface oxidation could be minimized. We have put detailed of device fabrication process in the Methods section of the revised main text.
- b) In Nat. Commun. 14: 2492, the authors found flake samples possess a weaker electron-phonon coupling compared with bulk crystals by analyzing thickness-dependent Raman response of CsV_3Sb_5 . An intuitively, such reduced interlayer coupling could suppress the 3D CDW ordering along the c axis. This scenario is supported by numerous studies in the broader field of 2D materials research, where the thinner the sample, the weaker the interlayer interactions (Nat. Commun. 9:1427). Owing to the competition between superconductivity and CDW, T_{CDW} decreases and T_c increases in thinner flakes. In our Raman measurements as shown in **Figure R6d**, it can be seen that the A_{1g} (out-of-plane vibration) mode in the flake sample undergoes a blue shift as compared to the bulk crystals, indicating that the flake sample has a weaker interlayer coupling (Nat. Commun. 14: 2492). Thus, our experiment is consistent with this picture.

Based on the above discussions, we believe that possible strain from the substrate in our samples is not related to the observed unusual anisotropic MR below $T^* \sim 35\text{K}$. To improve our manuscript, we added the related discussions on strain effects in the revised Supplementary Information S9 and S10, and added the discussions on origin of variation in T_c and T_{CDW} at Page 3, Line 90 in the revised main text.

- I cannot follow the symmetry arguments. A symmetry is either present or broken, and in this experiment it is throughout all parameter space explicitly reduced to C_2 , at all fields and temperatures. The Lorentz force owing to the authors choice of plane of rotation necessarily causes an asymmetry between transverse and longitudinal that arises purely from kinematics and is independent of the crystal symmetry of the probed solid. Once the symmetry of the data is lowered to C_2 , it is logically impossible to deduce a “further” symmetry lowering by fitting another $\cos(\theta)$ at a different phase and claiming a coexistence of multiple C_2 objects. The angle-dependence of MR is a very complex function, especially with complex Fermi surfaces and confined geometries / finite size effects. This complex object naturally can be decomposed into a Fourier series. The higher order terms of such decomposition correspond to structure beyond $\cos(\theta)$, and they fundamentally cannot be interpreted as an “additional” symmetry, such as here argued C_6 or a “second C_2 ”. The proper symmetry of this dataset necessarily is C_2 due to the chosen plane of rotation.

Reply: We are very grateful for the reviewer’s comments on the symmetry statements. We also thank the reviewer for pointing out the necessity to fully explain the symmetry argument, as it might not be straight-forward to readers. In this manuscript, all the discussion is about “symmetry broken” state, broken from the in-plane isotropic situation and reducing to new geometric symmetries. As the reviewer has pointed out, the key is: 1) whether MR could be a good tool to characterize these new symmetries; and 2) what physical origins of these new symmetries are, i.e., what causes the C_2^* , C_2 and C_6 rotational symmetries.

The effectiveness of the experimental and analysis method: We shall stress that magnetoresistance with rotating magnetic field and rotating current direction, as well as using sinusoidal functions to analyze the data, are broadly applied techniques in probing geometric symmetries of various electronic states. Previous works in the literature that applied this technique includes Nat. Phy. 8, 89; Nat. Phys. 12, 852; PRL 83, 2813; PRL129, 087002; PRX 8, 031002; Nat. Commun. 11: 874; Nat. Commun. 12, 6727; Nat. Commun. 14: 3046; Nat. Commun. 14: 7012; Nat. Commun. 14: 7155; Sci. Adv.2, e1601742, just to name a few. Thus, this method is solid. It is well accepted and regularly used by the community.

Will the first appeared symmetry always dominates: As commented by the reviewer, under small in-plane magnetic field, the classical Lorentz effect dominates. In this case, we are actually probing

the mild symmetry breaking caused by the application of a current for the transport measurement. However, as magnetic field increases, MR response from the nematicity as well as from the hidden phase of the sample becomes larger, which will naturally take over the classical effect and be the dominant source of the MR. Thus, the MR will appear to change shape, revealing the internal symmetry of the new dominant source of symmetry breaking. There are many such examples in the literature, with Nat. Commun. 14: 7155 being one of them.

The physical origin of the symmetry components: 1) The main finding of this manuscript is the C6 rotationally symmetric MR component that is from the hidden phase that appears at ~35K (Figure 2b in the main text), such C6 MR component is also accompanied by a strong upturn in the AHE signal (Figure 2d in the main text) as well as the appearance of a quasi-linear negative MR in both the $I//B$ and $I \perp B$ configurations (Figure 3e&3f in the main text) at around the same temperature. 2) A minor finding is the C2 MR component which is from the CDW phase, as well as with possible contribution from the electronic nematic phase (Nature 604, 59). In order to reach the above two findings, we also singled out the classical Lorentz force effect which is determined by the mild symmetry breaking by the application of the current (marked as C2*). All of the above has nothing to do with the actual symmetry of the kagome lattice itself. We can also exclude the possibility of the finite size effect since there are vast number of papers reported MR in different types of non-magnetic crystals with various thickness, but none of them observe such linear negative MR in thin flake samples. Should such linear negative MR arise from finite size effect, it should have been universally observed.

To reflect the comments of the reviewer, and make the description of MR symmetry more rigorous, we have appended the following clarification at Page 5, Line 118 in the revised manuscript: “Note that the above mentioned symmetries of the in-plane MR reflect the respective symmetry of various orders in CsV₃Sb₅ (see Supplementary Information S11), and can be characterized as various angular-dependent components of the in-plane MR^{24, 25}”

Minor points:

- Device fabrication. The discussed physics is bulk, and a discussion why these measurements were performed on flakes is lacking. Especially with all the issues from fabrication, it is strange this measurement was not done on macroscopic crystals. Please comment on the reason behind it and bind the fabrication more into the manuscript. The device fabrication and details (geometry?) are missing, Figure S1b lacks any scale information. Without a clear description on the fabrication steps it is very difficult to assess the work.

Reply: We appreciate the careful review and the helpful suggestions by the reviewer. Electrical transport measurements, unlike STM or other optical spectroscopic techniques that measure a specific microregion, provide information about the overall sample. However, the existence of

numerous domains in bulk samples can be disadvantageous for detecting the signal of symmetry breaking by electrical transport measurements. Recent scanning birefringence microscopy experiments (Nat. Phys., 18, 1470) confirmed the presence of numerous domains in CsV₃Sb₅ bulk samples, with domain sizes typically on the order of hundreds of micrometers. The devices measured in our studies are on the order of ten micrometers in size, which can effectively prevent the symmetry-breaking effects introduced by domains and can make the experimental results more reliable.

According to reviewer's suggestion, we have added a discussion on the size of devices at Page 3, Line 83 and more details of device fabrication in the **Methods** section in the revised main text, and added a scale bar in **Figure S1b** of the revised supplementary information.

- Unknown values have been subtracted from all data in the polar plots, making it impossible to understand the magnitude of the effect on the overall resistance. Please plot proper MR as a dimensionless quantity.

Reply: According to the reviewer's suggestions, we have plotted the angle dependent MR as a dimensionless quantity $(R_{xx}(B)-R_{xx}(0))/R_{xx}(0)$ in **Figure 1**, **Figure S4.1** and **Figure S4.2**. The scale bars in the figures now show the magnitude of the change in MR.

- What is the crystallographic alignment of the device? Is the C6 component in registry with the kagome net or rotated at an arbitrary angle.

Reply: In **Figure R8d** (copied from Figure R3 for the convenience of the reviewer), we plotted the angle dependent in-plane MR and kagome net whose orientation is determined by Raman measurements (Phys. Rev. Research 4, 023215; Nat. Commun. 14, 2492). The lobes of MR almost align with the symmetric axis of the kagome net. We have put the results in the revised Supplementary Information 9.

Figure R8. Orientation of the CsV₃Sb₅ flake sample detected by Raman spectra and in-plane MR. **a**, The optical picture of flake sample with Hall bar electrodes. **b**, Raman scattering intensity measured in parallel polarization with polarization rotated 180° on flake sample. A_{1g} and E_{2g} phonon frequency were observed. **c**, Polarization angle dependence of E_{2g} phonon intensities in flake sample. **d**, In-plane MR vs. kagome lattice. The orientation of kagome lattice is determined by Raman data in **c**. All the Data were collected at 5 K.

In conclusion, I cannot support the publication of the manuscript. It is unclear what exactly one learns from this experiment. The transport situation is extremely complex (finite size, complex Fermi surface, varying Lorentz force effects, substantial strain) so that quite naturally very complex patterns in the angle-dependent MR are to be expected and are observed. Disentangling them into microscopic insights is very difficult. Transport probes everything, hence it is not surprising that these experiments detect anomalies at T_{CDW}, T' and T_c; in accordance with the well established literature about them. It is difficult for me to pinpoint, setting the methodological issues aside, what new insights this work provides. A broken C₂ state at T' has already been seen by multiple groups, e.g. Y. Xiang et al., Nat. Comm. 12:6727 (2021), yet even that is difficult to state conclusively given that in the presented experiment the apparent symmetry is always C₂ even in higher symmetric metals such as copper.

Reply: Despite numerous studies into the hidden order at 35 K in CsV₃Sb₅, there is still no conclusion about it. STM, nuclear magnetic resonance (NMR), and elastoresistance measurement have pointed to the formation of electronic nematic order at and below ~35 K. However, nematicity can't explain the time reversal symmetry breaking in the crystal at around this temperature (arXiv: 2107.10714). Another candidate is the orbital current order, but its existence has also been disputed (PRB 105, 045102). Undoubtedly, the community needs a new angle to investigate this order. Our

results provide a deeper understanding of this hidden order via widely accepted and reliable transport method. The important findings are summarized as the following three points:

1. **First to discover unusual C6 component of the in-plane MR.** In addition to the C2 symmetry mentioned in reference Nat. Commun. 12:6727, we are the first to found that the in-plane MR contains a magnetic field-tunable C6 component that fades away above ~ 35 K. This observation indicates that the symmetry of the hidden phase below 35 K is six-fold, which is not yet reported by other groups. Combined with the time-reversal symmetry breaking, it points to one of the candidates, orbital current order with three-dimensional coupling. This finding indicates that both nematic order and orbital current order are involved in the transition at 35 K.
2. **First to show the three-dimension nature of the hidden order.** In further experiments for the revision, we find that the C2 and C6 components of in-plane MR are not present in samples with a thickness of ~ 20 nm. Our previous studies have shown that reducing the thickness of CsV₃Sb₅ causes a dimensional crossover from 3D to 2D to occur at a thickness below ~ 30 nm. This indicates that the hidden order is 3D in nature. In reference Nature 604, 59, the authors propose that nematic order can be regarded as a back-influence of the inter-layer coupling on the electronic properties within a single layer. But what is the inter-layer coupling is still unknown. **Our result is the first** to illustrate that the coupling of interlayer orbital current order is important in the intertwined orders in CsV₃Sb₅. We added this new evidence of 3D hidden phase at Page 11, Line 265 in the revised manuscript.
3. **First to report linear Negative in-plane MR in CsV₃Sb₅.** We find that the hidden order is always accompanied by a large linear negative in-plane MR that cannot be explained by conventional mechanisms, such as spin-dependent scattering, weak localization, and the effect of non-trivial band topology. Obviously, the negative MR observed here is related to the orbital current order. To explain this unusual negative MR, we proposed a phenomenological model: in-plane magnetic field suppresses fluctuations of the orbital current order, leading to weaker scattering and negative in-plane MR. This finding provides the potential of magneto-transport method to uncover orbital current order in other related materials.

We respectfully argue that the above three findings are not reported anywhere else yet and are highly valuable to the community for understanding exotic orders in kagome metals. Thus, we believe the readers of Nature Communications will benefit from learning the result of this work.

REVIEWER COMMENTS

Reviewer #1 (Remarks to the Author):

My concerns have been addressed, I would recommend publication of this work in Nature Communications.

Just a minor comment to the authors. In the rebuttal letter, there is one sentence said "our experiment served to provide a strong transport evidence of TRSB at ~35K, which would help to clarify the current debate." The transport results in this work show the evidence of a hidden phase around 35 K, but they cannot give strong evidence of TRSB.

Reviewer #2 (Remarks to the Author):

This is my second reading of the paper by X. Wei et al., who discuss transport signatures of the 35K transition in CsV₃Sb₅. I appreciate the authors work and effort in responding and revising the manuscript, however I find my points more concerning in this discussion and cannot support the publication.

1) Deviations of T_c and TCDW from the bulk values. Looking at the extremely enhanced values compared to the bulk, I assumed that strain is at the origin. I thank the authors for pointing me to the other two possibilities unique to nanoflakes, surface doping through oxidation and thickness-dependent electron-phonon coupling. I agree given the Raman measurements that no extensive strain is in the samples, while I am surprised by that finding. Nevertheless, the main issue stands: The electronic system of CVS is very sensitive, and something perturbed it as to change T_c by 72% as compared to the bulk! Lets assume that the authors hypothesis is correct, then this corresponds to a significant change of the electron-phonon interaction in thin flakes compared to bulk crystals. The conclusions drawn in this paper, however, claim to hold for bulk CVS which clearly is a very different electronic system. Given the sensitivity of CVS to perturbations, it is not at all clear that this response of a nanoflake is representative of the bulk system. In the way this manuscript is presented, the nanoflake is an unnecessary complication of the experiment. Beyond the discussions on how nanoflakes deviate from the bulk, the findings are directly ascribed as universal facts to CVS in any form. Just as an example, the abstract describes physics of CVS in general and the findings, without mentioning the issues of the substantial deviations of the nanoflake compared to the bulk. To be convincing that these effects are present in bulk, they should be performed in bulk crystals that follow the same physics as bulk crystals do.

2) This is simply not how symmetry works, and it is not how it is applied by the mentioned references. Once the symmetry is lowered to C_2 , there is no more meaning in terms of symmetry of a C_6 Fourier component. It appears very likely that the authors pick up on the electronic transition of the 35K phase, given the temperature-dependence of their data. However, the interpretation in terms of a state “characterized by six-fold rotational symmetry” is just not right.

Reviewer #3 (Remarks to the Author):

Report on "Three-dimensional hidden phase probed by in-plane magnetotransport in kagome metal CsV_3Sb_5 "

I have carefully reviewed the manuscript, the two referee reports from the previous reviewers, the responses from the authors, and the revised manuscript.

The manuscript reports comprehensive angular-dependent magnetoresistance (AMR) studies in the CDW ordered metallic state of the new kagome superconductor CsV_3Sb_5 . This is one of the fastest growing field of quantum materials because of the emergent rich and unusual correlated electronic states. The most important of these is the increasing evidence for time-reversal symmetry breaking and its interplay with the apparent rotation symmetry breaking. Understanding these novel symmetry breaking states is currently a high priority with far-reaching implications.

The authors studied the AMR as a function of temperature and magnetic field in thin film CsV_3Sb_5 and provided transport evidence for a hidden order phase below about 35K, consistent with the findings by other experimental probes reported previously. They further argue by symmetry analysis of the rotation symmetry breaking response that the observations support a three-dimensional time-reversal symmetry breaking orbital loop current order as the hidden order. The previous reviewers raised important questions and concerns on the validity of the interpretation. The responses by the authors are effortful and led to improvements of the manuscript, although in several places remained inconclusive because it is difficult at this stage to directly connect AMR to loop currents microscopically. I am not worried about how representative the thin film results are for bulk CsV_3Sb_5 , since their novelty stands alone and it is just as important to understand these fascinating physical behaviors in the thin films. I am also not too worried about making more precise and concrete connections to circulating loop current hidden order, since this is after all an experimental work and the authors are inspired and allowed to discuss the findings in this context so long as the discussions are scientifically sound. In fact, I find it valuable to put these extensive magnetotransport

studies including the anomalous Hall responses together and discuss the possible implications for orbital current order, which is currently lacking in the literature.

That said, I would like the authors to address the following issues, some of which are crucial no matter in which journal the manuscript is published. I will make my recommendation with regard to publication in Nature Communications based on the response from the authors.

(A) The introduction of the manuscript is poorly written. It is impossible for the reader to gain a reasonable understanding of the development in the field. The references are seriously incomplete and the descriptions are inaccurate and at places misleading. This must be improved for publication in any journal, but especially for a high impact journal like Nature Communications.

1. The authors must distinguish what are established and confirmed, what are evidenced but debated, and what are just proposals at the current stage. For example, unconventional superconductivity in Ref. 10 is merely a theoretical proposal and there is little experimental evidence pointing to non-phonon mediated superconductivity. The existence of time-reversal symmetry breaking is highly debated with conflicting experimental evidence. The proposal for chiral charge order is under scrutiny amid these debates, albeit if time-reversal symmetry is indeed broken, then circulating orbital loop current order is a likely microscopy origin. The statement for theoretically predicted and experimentally confirmed properties simply cannot include chiral 3D charge order, Majorana zero mode, and orbital current order.

2. The references for time-reversal symmetry breaking are incomplete. The authors cited the muSR papers (an important one is missing: Khasanov, et. al. PRR 4, 023244 (2022)), but omitted the optical kerr rotation and circular dichroism, with only Ref. 27 mentioned later and Ref.20 added in the later part for the domain size. These crucial experiments directly probing time-reversal symmetry breaking as well as rotation symmetry breaking should be referred to in the introduction together.

Ref. 27,

Ref. 20,

Saykin et al, PRL 131, 016901 (2023)

Hu et al, arXiv.2208.08036

Farhang et al, Nat. Commun. 14, 5326 (2023)

3. It is misleading to cite only Refs. 15 and 16 for the theoretical proposal of orbital loop-current order. There are several original theoretical works that have been omitted with some more relevant to the current experiments, e.g.

Denner et al, PRL 127, 217601 (2021)

Park et al, PRB 104, 035142 (2021)

Lin et al, PRB 104, 045122 (2021)

Zhou et al, Nat. Commun. 13, 7288 (2022)

Christensen et al, PRB 106, 144504 (2022)

4. Rotation symmetry breaking CDW order and the appearance of rotational symmetry breaking coherent quasiparticles in the $\sim 35\text{K}$ phase have been reported by STM in Nat. Phys. 19, 637 (2023), which is directly relevant for this work. It should be mentioned or discussed in connection to the current findings.

(B) AMR is a technique aimed for studying rotation symmetry breaking and does not probe directly time-reversal symmetry breaking. It is thus not straightforward to invoke orbital current order to account for some of the key observations, unless other simpler possibilities have been ruled out or argued scientifically to be unlikely. Given that the crystalline rotation symmetry has been broken down to C_2 , one must ask whether some of the observed phenomena are due to the contributions from the three possible C_2 domains. This would seem natural to account for the differently lobed AMR polar plots as a function of temperature and magnetic field. The authors argue that the optical kerr rotation experiments (Ref.20) found C_2 domains of hundreds of microns in size, which is much larger than the measurement dimension of the thin film devices. This argument is weak, to say the least. These are large domains on the length scale relevant for the optical probes in bulk sample. Even for bulk sample, 120° degree oriented C_2 domains have been observed by STM on much smaller length scales (Li et al, Nat. Phys. 18, 265 (2022), etc.). It is thus very likely that domains of much smaller sizes exist even in bulk samples. In thin films, these domains can be more proliferate. Thus it is important that the authors address whether the domain contributions on the scale comparable or smaller than the measurement thin film devices can contribute to the observed properties. In the event that these cannot be ruled out, the authors ought to tune down the claim for orbital current order and include domain contributions as a possibility.

(C) The manuscript can benefit substantially from discussions related to the more recent advances in the field on the nature of the CDW state. Recently, laser and magnetic field STM experiments (Xing et al, arXiv:2308.04128) reported that the different CDW peak heights detected by STM in Ref. 4 is directly connected to the lattice distortions. This can be induced due to electron-phonon coupling.

The time-reversal symmetry breaking determined by the magnetic field response is also accompanied by the lattice distortions in response to flipping the magnetic field along the c-axis. These are relevant for the current work. In particular, a specific rotation symmetry breaking orbital current state was proposed in that work. It would increase the timeliness and the scientific level of the current manuscript if discussions were provided in view of these latest developments.

Reviewer #1 (Remarks to the Author):

My concerns have been addressed, I would recommend publication of this work in Nature Communications.

Just a minor comment to the authors. In the rebuttal letter, there is one sentence said “our experiment served to provide a strong transport evidence of TRSB at ~35K, which would help to clarify the current debate.” The transport results in this work show the evidence of a hidden phase around 35 K, but they cannot give strong evidence of TRSB.

Reply: We are glad that the reviewer considers our responses satisfactory and recommends the publication of our current manuscript in Nature Communications. We appreciate the reviewer’s comments that help us greatly improve our manuscript.

In addition, we fully agree with the reviewer's rigorous comment that our transport experiment cannot directly demonstrate the occurrence of time-reversal symmetry breaking (TRSB) at 35K. Our work suggests the presence of a hidden phase transition at this temperature. Combined with other experimental and theoretical articles, this hidden phase transition appears to be associated with the occurrence of a three-dimensional orbital current order. Thus, we would like to modify our response to: “Our findings offer a plausible mechanism to account for the enhanced signal or emergence of TRSB observed at 35K through μ SR and chiral transport experiments.” Since this particular sentence only appeared in the previous round of rebuttal letter, and not in the revised manuscript, we didn’t make additional modification to the manuscript.

Reviewer #2 (Remarks to the Author):

This is my second reading of the paper by X. Wei et al., who discuss transport signatures of the 35K transition in CsV_3Sb_5 . I appreciate the authors work and effort in responding and revising the manuscript, however I find my points more concerning in this discussion and cannot support the publication.

Reply: We appreciate the reviewer for his/her detailed comments on our revised manuscript. We have noticed that the reviewer's main concerns lie in: 1) the different properties between nanoflake and the bulk, and 2) the physical interpretation of the C6 components. In the following, we provide comprehensive responses to these concerns, aiming to address the reviewer's questions.

1) Deviations of T_c and T_{CDW} from the bulk values. Looking at the extremely enhanced values compared to the bulk, I assumed that strain is at the origin. I thank the authors

for pointing me to the other two possibilities unique to nanoflakes, surface doping through oxidation and thickness-dependent electron-phonon coupling. I agree given the Raman measurements that no extensive strain is in the samples, while I am surprised by that finding. Nevertheless, the main issue stands: The electronic system of CVS is very sensitive, and something perturbed it as to change T_c by 72% as compared to the bulk! Lets assume that the authors hypothesis is correct, then this corresponds to a significant change of the electron-phonon interaction in thin flakes compared to bulk crystals. The conclusions drawn in this paper, however, claim to hold for bulk CVS which clearly is a very different electronic system. Given the sensitivity of CVS to perturbations, it is not at all clear that this response of a nanoflake is representative of the bulk system. In the way this manuscript is presented, the nanoflake is an unnecessary complication of the experiment. Beyond the discussions on how nanoflakes deviate from the bulk, the findings are directly ascribed as universal facts to CVS in any form. Just as an example, the abstract describes physics of CVS in general and the findings, without mentioning the issues of the substantial deviations of the nanoflake compared to the bulk. To be convincing that these effects are present in bulk, they should be performed in bulk crystals that follow the same physics as bulk crystals do.

Reply: We thank the referee for his/her rigorous comments. In our response to the first round of reviewer comments, we have provided the reasons for not measuring bulk samples: owing to the existence of numerous domains in bulk materials, electron scattering from domains in MR would obscure that from the various ordered states, thus rendering it difficult to obtain effective information of ordered states from MR. In order to be precise and to reflect the reviewer's comment, we have changed the title of this manuscript to be: "Three-dimensional hidden phase probed by in-plane magnetotransport in kagome metal CsV_3Sb_5 thin flakes". We have also added the phrase "thin flakes" in the abstract of the manuscript: "Here we present magneto-transport evidence of a new phase below ~ 35 K in the kagome topological metal CsV_3Sb_5 (CVS) thin flakes between the CDW and the superconducting transition temperatures." We have also rephrased the relevant descriptions in the revised main text, emphasizing the thin-film samples used in our studies. We hope such modification could precisely convey the idea that we have obtained such research in thin flake samples. Given the rephrasing, we would like to stress that it does not in any way diminish the significance and novelty of our research, as mentioned by reviewer #3.

2) This is simply not how symmetry works, and it is not how it is applied by the mentioned references. Once the symmetry is lowered to C_2 , there is no more meaning in terms of symmetry of a C_6 Fourier component. It appears very likely that the authors

pick up on the electronic transition of the 35K phase, given the temperature-dependence of their data. However, the interpretation in terms of a state “characterized by six-fold rotational symmetry” is just not right.

Reply: We agree with the reviewer that, from the perspective of global symmetry, once lowered to C2, there is no way to increase the global symmetry with additional electronic states that have higher symmetry. However, our study did not focus on the global symmetry that include all the phases of the entire system, but rather, looking into the respective symmetries of different phases, which gives important information of those particular phases. The intricate intertwining and coupling of various phases in kagome metal CVS leads to complex electronic behaviors in the system. To separate and identify these phases is undoubtedly of great scientific significance. Fortunately, we found that the different phases exhibit distinct magnetic field and temperature dependent responses, that help us distinguish the signal from trivial origins.

We are confident that the C6 component cannot be the higher order Fourier terms (we shall call it “higher harmonic terms” later in the discussion) of the C2 component from the following three reasons: 1) if the main symmetry is C2, the amplitude of the higher-harmonic terms, including the C6 terms and other higher-harmonic terms such as C8, C10, C12, should be much smaller than the amplitude of the C2 term, which is not observed in our experiment; 2) there should be other higher harmonic terms, e.g., C8, C10, C12, etc., if they are simply mathematical glitches, which are also not presented; 3) since these higher harmonic terms have the same physical origin as the C2 term, they should behave in a similar way when temperature and magnetic field changes, which is again not the case in our experiment.

That being said, the reviewer is correct that higher harmonic terms might be a factor of the observed signal, and we cannot completely rule out the possibility that a small portion of the C6 signal would have come from the higher harmonic term. Thus, to reflect the comment of the reviewer, we have modified the main text at page 5, line 126 to be: “Note that the above-mentioned symmetries of the in-plane MR reflect the respective symmetry of various orders in CsV₃Sb₅ (see Supplementary Information S11), and can be characterized as various angular-dependent components of the in-plane MR^{24,25}. We also note that higher-harmonics of the C2 term with small amplitudes cannot be completely ruled out, which might exhibit as C2n terms, where n = 2, 3, 4, 5..., and these higher-harmonic terms should behave similarly to the C2 term when subject to changes in external magnetic field and temperature due to the fact that they have the same physical origin.”

Reviewer #3 (Remarks to the Author):

Report on "Three-dimensional hidden phase probed by in-plane magnetotransport in kagome metal CsV_3Sb_5 "

I have carefully reviewed the manuscript, the two referee reports from the previous reviewers, the responses from the authors, and the revised manuscript.

The manuscript reports comprehensive angular-dependent magnetoresistance (AMR) studies in the CDW ordered metallic state of the new kagome superconductor CsV_3Sb_5 . This is one of the fastest growing field of quantum materials because of the emergent rich and unusual correlated electronic states. The most important of these is the increasing evidence for time-reversal symmetry breaking and its interplay with the apparent rotation symmetry breaking. Understanding these novel symmetry breaking states is currently a high priority with far-reaching implications.

The authors studied the AMR as a function of temperature and magnetic field in thin film CsV_3Sb_5 and provided transport evidence for a hidden order phase below about 35K, consistent with the findings by other experimental probes reported previously. They further argue by symmetry analysis of the rotation symmetry breaking response that the observations support a three-dimensional time-reversal symmetry breaking orbital loop current order as the hidden order. The previous reviewers raised important questions and concerns on the validity of the interpretation. The responses by the authors are effortful and led to improvements of the manuscript, although in several places remained inconclusive because it is difficult at this stage to directly connect AMR to loop currents microscopically. I am not worried about how representative the thin film results are for bulk CsV_3Sb_5 , since their novelty stands alone and it is just as important to understand these fascinating physical behaviors in the thin films. I am also not too worried about making more precise and concrete connections to circulating loop current hidden order, since this is after all an experimental work and the authors are inspired and allowed to discuss the findings in this context so long as the discussions are scientifically sound. In fact, I find it valuable to put these extensive magnetotransport studies including the anomalous Hall responses together and discuss the possible implications for orbital current order, which is currently lacking in the literature.

That said, I would like the authors to address the following issues, some of which are crucial no matter in which journal the manuscript is published. I will make my recommendation with regard to publication in Nature Communications based on the response from the authors.

Reply: We would like to thank the reviewer for carefully reviewing our manuscript, the two referees' reports from the previous reviewers, our responses and the revised manuscript, and for providing rigorous and objective comments. We also appreciate the reviewer's recognition of the scientific value of our findings and his/her affirmation of our efforts to improve the manuscript. In this revision, we hope that we have addressed the relevant issues raised by the reviewer.

(A) The introduction of the manuscript is poorly written. It is impossible for the reader to gain a reasonable understanding of the development in the field. The references are seriously incomplete and the descriptions are inaccurate and at places misleading. This must be improved for publication in any journal, but especially for a high impact journal like Nature Communications.

Reply: We thank the reviewer for pointing out the issues of introduction and references, that help us substantially improve our manuscript. According to the reviewer's suggestions and comments, we have modified the introduction and references, the details as listed in the following.

1. The authors must distinguish what are established and confirmed, what are evidenced but debated, and what are just proposals at the current stage. For example, unconventional superconductivity in Ref. 10 is merely a theoretical proposal and there is little experimental evidence pointing to non-phonon mediated superconductivity. The existence of time-reversal symmetry breaking is highly debated with conflicting experimental evidence. The proposal for chiral charge order is under scrutiny amid these debates, albeit if time-reversal symmetry is indeed broken, then circulating orbital loop current order is a likely microscopy origin. The statement for theoretically predicted and experimentally confirmed properties simply cannot include chiral 3D charge order, Majorana zero mode, and orbital current order.

Reply: We fully agree with the reviewer's rigorous comment that the concept "unconventional superconductivity" in this system is merely a theoretical proposal without experimental evidence. Indeed, we should be very cautious to use this term, because there are still ongoing controversies even in copper oxide superconductors. We have also noticed that recent researches offer different perspectives on spontaneously time-reversal symmetry breaking in the system. To account for these new insights, we have included additional statements and corresponding references at the second paragraph of the introduction (marked in blue at Page 3, Line 67). Additionally, we thank the reviewer for pointing out the misleading or inaccurate statement regarding

orbital current order, chiral 3D charge order and Majorana zero mode. We have substantially modified the first paragraph of introduction, which is marked in blue in the revised manuscript, at Page 2, Line 51: “The recently discovered kagome topological metal AV_3Sb_5 ($A=Cs, Rb, K$) has proven to be a valuable material platform for studying topological states and electron correlations^{1,2,3,4,5,6,7,8}. It features a wealth of states of matter and interesting electronic behaviors, including topological surface states^{2,9}, superconductivity with pair density wave³, electronic nematicity⁸, charge density wave⁴, chiral transport¹⁰, anomalous Hall effect¹¹ and time-reversal symmetry breaking⁷, among others. Such intricate and diverse range of states has sparked great interest, and numerous experiments are quickly focused on the search for potentially impactful quantum states within this system, such as unconventional superconductivity^{4,12,13,14}, Majorana zero mode^{5,15}, and orbital current order^{16,17,18,19,20,21,22}.”

We have also added a discussion at Page 3, Line 67 in the revised manuscript: “Meanwhile, another STM experiment²⁶ found that the unidirectional coherent quasiparticles appear below 30 K. These studies altogether presented a puzzling physical picture, that the hidden phase below ~ 35 K simultaneously breaks the rotational symmetry and time-reversal symmetry. Moreover, its mechanism become more confusing, since different conclusions have been reported recently, that spontaneously time-reversal symmetry breaking either coincides with CDW^{27,28,29} or it does not occur at all^{30,31}, and rotational symmetry breaking also occurs at higher temperatures^{26,27,28}.”

2. The references for time-reversal symmetry breaking are incomplete. The authors cited the muSR papers (an important one is missing: Khasanov, et. al. PRR 4, 023244 (2022)), but omitted the optical kerr rotation and circular dichroism, with only Ref. 27 mentioned later and Ref.20 added in the later part for the domain size. These crucial experiments directly probing time-reversal symmetry breaking as well as rotation symmetry breaking should be referred to in the introduction together.

Ref. 27,

Ref. 20,

Saykin et al, PRL 131, 016901 (2023)

Hu et al, arXiv.2208.08036

Farhang et al, Nat. Commun. 14, 5326 (2023)

Reply: We appreciate the helpful suggestion of the reviewer. We have added the reference at Page 3, Lines 63, 71 and 72 in the revised manuscript.

3. It is misleading to cite only Refs. 15 and 16 for the theoretical proposal of orbital loop-current order. There are several original theoretical works that have been omitted with some more relevant to the current experiments, e.g.

Denner et al, PRL 127, 217601 (2021)

Park et al, PRB 104, 035142 (2021)

Lin et al, PRB 104, 045122 (2021)

Zhou et al, Nat. Commun. 13, 7288 (2022)

Christensen et al, PRB 106, 144504 (2022)

Reply: We thank reviewer for the careful comments and for pointing out the incomplete citations. We have added the relevant references at Page 2, Line 58 in the revised manuscript.

4. Rotation symmetry breaking CDW order and the appearance of rotational symmetry breaking coherent quasiparticles in the $\sim 35\text{K}$ phase have been reported by STM in Nat. Phys. 19, 637 (2023), which is directly relevant for this work. It should be mentioned or discussed in connection to the current findings.

Rely: We thank the referee for his/her helpful suggestions. We have added this reference and its discussion at Page 3, Line 67 in the revised manuscript: “Meanwhile, another STM experiment²⁶ found that the unidirectional coherent quasiparticles appear below 30 K. These studies altogether presented a puzzling physical picture, that the hidden phase below $\sim 35\text{ K}$ simultaneously breaks the rotational symmetry and time-reversal symmetry.”

(B) AMR is a technique aimed for studying rotation symmetry breaking and does not probe directly time-reversal symmetry breaking. It is thus not straightforward to invoke orbital current order to account for some of the key observations, unless other simpler possibilities have been ruled out or argued scientifically to be unlikely. Given that the crystalline rotation symmetry has been broken down to C_2 , one must ask whether some of the observed phenomena are due to the contributions from the three possible C_2 domains. This would seem natural to account for the differently lobed AMR polar plots as a function of temperature and magnetic field. The authors argue that the optical kerr rotation experiments (Ref.20) found C_2 domains of hundreds of microns in size, which is much larger than the measurement dimension of the thin film devices. This argument is weak, to say the least. These are large domains on the length scale relevant for the optical probes in bulk sample. Even for bulk sample, 120 degree oriented C_2 domains have been observed by STM on much smaller length scales (Li et al, Nat. Phys. 18, 265

(2022), etc.). It is thus very likely that domains of much smaller sizes exist even in bulk samples. In thin films, these domains can be more proliferate. Thus it is important that the authors address whether the domain contributions on the scale comparable or smaller than the measurement thin film devices can contribute to the observed properties. In the event that these cannot be ruled out, the authors ought to tune down the claim for orbital current order and include domain contributions as a possibility.

Reply: First, we agree with the reviewer that our experimental results provide direct information for rotational symmetry, not for time-reversal symmetry; the discussion on the possibility of a three-dimensional orbital current order is a proposal with the current evidence from this experiment and from the literature. Actually, the statement about time-reversal symmetry breaking is based on previous reports, such as μ SR experiment and optical Kerr experiment.

Second, we thank the reviewer for bringing up a new possible mechanism for the C6 component. There exist three C2 domains with angles of 120° among them, each of which contributes to a lobe of in-plane MR, forming C6 component. Indeed, it is very difficult for us to directly rule out the existence of domains. However, it is well known that electron scattering by each domain in MR should be uniform. Then assuming this proposal is correct, the magnetic field and temperature dependence of C2 and C6 ought to exhibit comparable behavior (due to the fact that they are all from C2 domains according to this scenario), which is inconsistent with our experimental results. Therefore, we believe that C6 component of MR in our work does not originate from C2 domains, or, at least, the dominant C6 signal is not from C2 domains. In the context of current researches, orbital current order could stand as the most plausible candidate mechanism for the hidden order, despite the lack of conclusive evidence to support its existence. To reflect the comments of the reviewer, we have added the discussion on domains at Page 7, Line 179 in the revised manuscript: “This C6 component involves three potential origins: 1) the symmetry of the kagome lattice, 2) three C2 order domains with angles of 120° between them, and 3) the symmetry of a possible orbital current order. The first mechanism can be easily excluded since the kagome lattice structure is unlikely to be altered by an in-plane magnetic field of ~ 10 T. For the second scenario, it is very difficult for us to directly examine the existence of domains. However, electron scattering rate by each domain per unit area should be uniform and they contribute the MR of the devices. Then assuming the second scenario is correct, the magnetic field and temperature dependence of C2 and C6 ought to exhibit comparable behavior (due to the fact that they are all from C2 domains), which is inconsistent with our experimental results. Therefore, we believe that C6 component of MR does not originate from C2 domains, or, at least, the dominant C6 signal is not from

C2 domains. In the context of current researches, orbital current order could stand as the most plausible candidate mechanism for the C6 component of the MR.”

(C) The manuscript can benefit substantially from discussions related to the more recent advances in the field on the nature of the CDW state. Recently, laser and magnetic field STM experiments (Xing et al, arXiv:2308.04128) reported that the different CDW peak heights detected by STM in Ref. 4 is directly connected to the lattice distortions. This can be induced due to electron-phonon coupling. The time-reversal symmetry breaking determined by the magnetic field response is also accompanied by the lattice distortions in response to flipping the magnetic field along the c-axis. These are relevant for the current work. In particular, a specific rotation symmetry breaking orbital current state was proposed in that work. It would increase the timeliness and the scientific level of the current manuscript if discussions were provided in view of these latest developments.

Reply: We appreciate the great suggestion of the reviewer. We carefully read the paper (Xing et al, arXiv:2308.04128), in which they utilized STS technique and found that the so-called chirality or time reversal symmetry breaking of charge order was determined by electric or magnetic field induced in-plane lattice distortions. Meanwhile they proposed a simple CDW model, namely out-of-phase combination of bond charge order and loop currents. This study provides a nice angle to look into the possible time reversal symmetry breaking and rotational symmetry breaking in AVS, which effectively reconciles previous contradictions in the literature, and provides information on the specific orbital current state associated with rotation symmetry breaking. The STM study is in line with our quantum transport work and provides a compelling explanation for the concurrent transitions of C2 and C6 components observed near 35K in our experiment. In comparison to this STM study, our transport data further reflects the response of the electronic state to the in-plane magnetic field, and illustrates the three-dimensional properties of orbital current order. We have added the discussion about this reference at Page 9, Line 219 in the revised manuscript: “In addition, we noticed that the concurrent transitions of C2 and C6 components observed near 35 K can be explained by the out-of-phase combination of bond charge order and orbital current order, as proposed in a more recent STM study³⁹.”

To summarize, we would like to thank all the reviewers for their rigorous comments and helpful suggestions, and they really contributed to a much better version of the manuscript that can bring more value and information to the readers of our paper.

REVIEWERS' COMMENTS

Reviewer #3 (Remarks to the Author):

I have reviewed the responses from the authors and the revised manuscript. The authors have addressed all my criticisms, comments and suggestions. The manuscript has been improved significantly and is more solid and more rigorous scientifically. I recommend publication in Nature Communications.

Point-by-point response:

Reviewer #3 (Remarks to the Author):

I have reviewed the responses from the authors and the revised manuscript. The authors have addressed all my criticisms, comments and suggestions. The manuscript has been improved significantly and is more solid and more rigorous scientifically. I recommend publication in Nature Communications.

Response: Thanks a lot for the brief and positive comment from Reviewer #3. We highly appreciate the contributions from all the reviewers that have improved our manuscript substantially. According to the current comment, we have kept the manuscript as is with no changes, except for the response to the editorial requests.